# Trend and factors associated with anemia among women reproductive age in Ethiopia: A multivariate decomposition analysis of Ethiopian Demographic and Health Survey

**Berhan Tsegaye Negash**[1]*, **Mohammed Ayalew**[2]

**1** Department of Midwifery, College of Medicine and Health sciences, Hawassa University, Hawassa, Ethiopia, **2** Department of Psychatric Nursing, College of Medicine and Health Science, Hawassa University, Hawassa, Ethiopia

* birieman67@gmail.com

**Data Availability Statement:** All relevant data are within the paper and its Supporting Information files.

## Abstract

### Background

In developing countries like Ethiopia, anemia is a public health problem. Unfortunately, the progress of anemia reduction has been slow. Although the issue of anemia has received considerable critical attention nowadays, trends and factors associated with anemia among women of reproductive age have not been explored in Ethiopia.

### Objective

This study aimed to determine trends and factors associated with anemia among women of reproductive age in Ethiopia from 2005 to 2016.

### Method

Data from three consecutive Ethiopian Demographic and Health Survey (EDHS) from 2005–2016 were analyzed in this study. EDHS is a two-stage cluster sampling survey. Data were weighted to correct sampling bias in all surveys. A total of 46,268 samples were analyzed using a fixed effect model. For a measure of proportion, differences and slopes were computed. Bivariate and multivariable logistic regression analyses were done to identify predictors of the trend of anemia among women. Adjusted odds ratio (AOR) with a 95% Confidence Interval(CI) was computed, and the p-value < 0.05 is considered significant.

### Result

Prevalence of anemia among women was 68%, 20.3%, and 27.3% in 2005, 2011 and 2016, respectively. The trend of anemia was reduced by 47.7 percentage points from 2005 to 2011; however, it increased by 7% points again from 2011 in 2016. Lack of mobile phones (AOR = 1.4, 95%CI, 1.2,1.6), Afar women (AOR = 1.5, 95%CI, 1.1,2.3) and Somali women (AOR = 1.5, 95%CI, 1.1,1.9) were associated with anaemia among women. On the contrary, the history of heavy menstruation in the last six months (AOR = 0.9; 95%CI, 0.85,0.98) was a factor negatively associated with anemia in 2005. In 2011, single women (AOR = 0.8,95%

**Funding:** The author(s) received no specific funding for this work.

**Competing interests:** The authors have declared that no competing interests exist.

CI,0.7,0.9), watching TV less than once per wk (AOR = 0.9,95%CI,0.7,0.95), watching TV at least once per week (AOR = 0.8,95%CI,0.7,0.98) were variables associated with anemia. On the contrary, widowed women (AOR = 1.7,95%CI,1.4,2.0) were affected by anemia. In 2016, the richest women (AOR = 0.7, 95%CI, 0.6,0.8) and single (AOR = 0.8, 95%CI, 0.7,0.9) were affected little by anemia. Women of traditional belief followers (AOR = 2.2,95%CI,1.6,2.9) were more highly influenced by anemia than their counterparts.

## Conclusion

The prevalence of anemia declined rapidly from 2005 to 2011, and increased from 2011 to 2016. Stakeholders should develop policies and programs to enhance the socio-demographic status of women and basic infrastructure for the community. Furthermore, they should design strategies for extensive media coverage of the prevention of anemia. The federal government should balance the proportion of anemia among regions by ensuring health equality.

## Background

Anemia is described as a state of reduction in the number of red blood cells or their oxygen-carrying capacity so that they can not supply the body's physiologic needs adequately [1]. It is characterized by fatigue, weakness, decreased productivity, and inhibited immune function [2, 3]. It can be caused by infections, chronic inflammatory conditions, micronutrient deficiencies or genetic disorders [4, 5]. Although anemia is caused by these etiologies, the most commonest (60%) type of anemia is iron deficiency anemia [6]. The causes of anemia vary by location, demography [6] or economic growth [7].

Globally, the prevalence of anemia varies across countries. For example, in the overall world population, nearly 14% of anemia occurs in high-income countries, and 50% of low-income countries are affected by anemia [8–10]. The world Health Organization (WHO) reported that women of reproductive age are at high risk of developing iron deficiency anemia [11]. Particularly, women in resource- poor countries often fail to get extra food and supplements needed to meet the necessary daily requirements for normal physiology during reproduction [12].

WHO classifies anemia based on the hemoglobin level as sever,moderate and mild anemia if its prevalence accounts for 40%, 20%-40%, and 5%-20%, respectively [13]. In Ethiopia, a meta-analysis study indicated that pooled prevalence of anemia was 31.6% among pregnant women [14]. Previous studies have indicated several predictors of anemia among women of reproductive age. These include the following socio-economic and reproductive health characteristics:young age, grand multi-parity, short inter-pregnancy intervals, low socio-economic status, low educational status, ignorance [15], heavy menstrual blood loss, history of using an intrauterine contraceptive device, previous history of anemia, and body mass index [16]. Anemia causes adverse pregnancy outcomes: still birth, abortion, low birth weight, maternal and neonatal mortality [17, 18], intrauterine fetal death, premature birth [19], decreases physical performance, and work capacity [20]. Consequently, the world Health Assembly (WHA) adopted a commitment to reduce the magnitude of anemia by half among women of reproductive age by 2025 [21].

According to the Ethiopian National Food Consumption Survey (ENFCS), dietary iron intake was high at the population level in Ethiopia, especially among adult women [22]. For

example, 64 percent of women of reproductive age had intakes that exceeded the World Health Organization's recommended upper intake levels [23]. Some techniques stated in the national nutrition program to avoid iron deficiency anemia among Ethiopian women of reproductive age include iron-folic acid supplementation, food fortification, and dietary diversification [24]. Pregnant women are given iron-folic acid tablets, but compliance is poor overall, ranging from 3.5% to 76% [25, 26]. Therefore, a food fortification strategy should be adopted. However, anemia is more prevalent among women of reproductive age. Even with the existence of a number of policies and programs, anemia among reproductive age women persists as a critical concern for the public health agenda [27]. Furthermore, previous studies have identified some key independent variables associated with anemia, such as residence, wealth index, and modern contraceptive use. However, the majority of them were small-scale, and conducted in specific localities rather than country-wide [28–31]. Reducing the prevalence of anemia among women of reproductive age is considered a vital strategy to improve the health of women at reproductive age. Consequently, the world health organization has set a global target of achieving a 50% of reduction in the proportion of anemia among women reproductive age by 2025 [32].

In Ethiopia, previous studies focused on the level of anemia in certain vulnerable population groups, such as children, pregnant, and lactating women. However, few studies were conducted on the prevalence of anemia among women of reproductive-age [30, 31, 33]. Furthermore, although most studies were conducted on anemia among reproductive age women in different local areas in Ethiopia, they fail to show trends and associated factors with anemia among women of reproductive age at the national level. Therefore, this study aimed to assess trends and factors associated with anemia among reproductive-age women in Ethiopia from 2005–2016.

## Methods

### Study setting and period

Ethiopia is one of the East African countries which had an estimated population of around 107,406,158 in 2017. Accordingly,Ethiopians make up 1.4% of the global population, putting in the 12[th] place worldwide. Administrative, Ethiopia is divided into 10 regions. Each region is divided into zones. Furthermore,zones are further sub-divided into districts. Finally, each district in turn is sub-divided into sub-districts or Kebeles. Kebeles are the lowest administrative units, which consist of around 1000 households. Ethiopia has a diversified population with more than 80 ethnic groups. Most (80%) of the population of Ethiopians live in rural areas [12, 13]. Based on the 2007 Ethiopian national census report, the average size of household was 5, 4.6, and 4.6 persons in 2005, 2011 and 2016 respectively. The population proportion of males to females is equivalent in Ethiopia. The total fertility rate has declined from 5.5 in 2000 to 4.1 in 2014. In Ethiopia, women of the reproductive age group constitute 23.4% of the total population [34, 35].

### Data availability

Ethiopian Demographic and Health Survey (EDHS) is a quantitative cross-sectional study conducted every five years in Ethiopia. The current research is a secondary data analysis of EDHS 2005,2011, and 2016. The main objective of DHS is to provide critical information on indicators of fertility, family planning, infant health, child health, adult health, maternal and child health, nutrition and sexually transmitted infections. First, we were registered and requested EDHS data for analysis from the 'measure DHS' online archive. Then, permission to access the database was officially obtained for this study. The database is available at the official website of DHS, which is found at the following link: https://dhsprogram.com.

## Study population

Women of reproductive age in Ethiopia from 2005 to 2016 (15–49 years) were the target population. Women who lived in the selected clusters and presented during data collection period were the study population in this study. Pregnant and smoker women were excluded in this study as cut off anemia threshold varies compared to women of reproductive age. The primary sampling unit was enumeration area/ cluster/. On the contrary, secondary sampling units were households in this study. Each woman was the study unit in this study.

## Sample size determination and sampling technique

The sample sizes for the EDHS are determined through consultation with senior statisticians, who consider survey precision, budget and accuracy. The Ethiopian central statistical agency conducts the Ethiopian national census. The 1994 Ethiopian national census was the source of sampling frame for EDHS 2005. Furthermore, the 2007 Ethiopian National Census was also the source of the sampling frame for EDHS 2011 and EDHS 2016. Enumeration areas(EAs) were created during the national population census.

Fig 1 displays that The EDHS is a stratified sampling technique survey with two stages. First, clusters were selected using a simple random sampling method. A total of 540 enumeration areas were used in the EDHS 2005. Furthermore, 624 clusters were considered for EDHS 2011, and 645 clusters were taken into account for EDHS 2016. In each cross-section of surveys, EAs were stratified into urban and rural areas. DHS is a two stage survey. In the first stage, clusters were selected and stratified by residence status. Households were chosen in the second stage in proportion to the number of households in each cluster. Each household was selected using systematic random sampling. Simple random sampling was applied to select women, if there were more than one eligible woman in the same household. The detailed methodology is explained in the survey reports [36]. In the first stage, clusters were divided into urban (145) and rural (395) clusters in the 2005 EDHS. Furthermore, in the 2011 EDHS, 187 urban and 395 rural clusters were considered. Finally, in the EDHS 2016, 202 urban and 443 rural clusters were used as the primary sampling units. Second, households were selected as the secondary sampling units from selected clusters proportionally. Accordingly, a total of 14,645 households were selected during 2005 EDHS. Besides, a total of 18,817 households were drawn from the sampling frame of clusters in 2011 EDHS. Lastly, a total of 18,008 households were used in EDHS 2016. However, among the selected households, only 13,181 households were occupied by people. Moreover, women lived in only 16,702 households in 2011 EDHS. Finally, nearly 16,650 households were occupied by women. Hence, the response rate of these surveys was considered as good in all three surveys. As a result, the response rate was reported as 99%,98.1% and 98% in 2005,2011 and 2016 respectively at the household level. Although these households were occupied by women in three consecutive surveys, all women were not eligible for these surveys. Hence, only 14,717 women were eligible in 2005. In addition, 7,385 women were eligible for EDHS 2011, but only 16,650 were eligible for EDHS 2016. However, only fourteen thousand seventy (14,070), sixteen thousand five hundred five-five (16,515), and fifteen thousand six hundred eighty-three (15,683) women have given full responses to questions, making a final response of 96, 95 and 95% during 2005 EDHS,2011EDHS, and 2016 EDHS, respectively (**Fig 1**).

## Operational definition

**Denominator.** Number of women age 15–49 years measured for anemia in households selected for anemia testing.

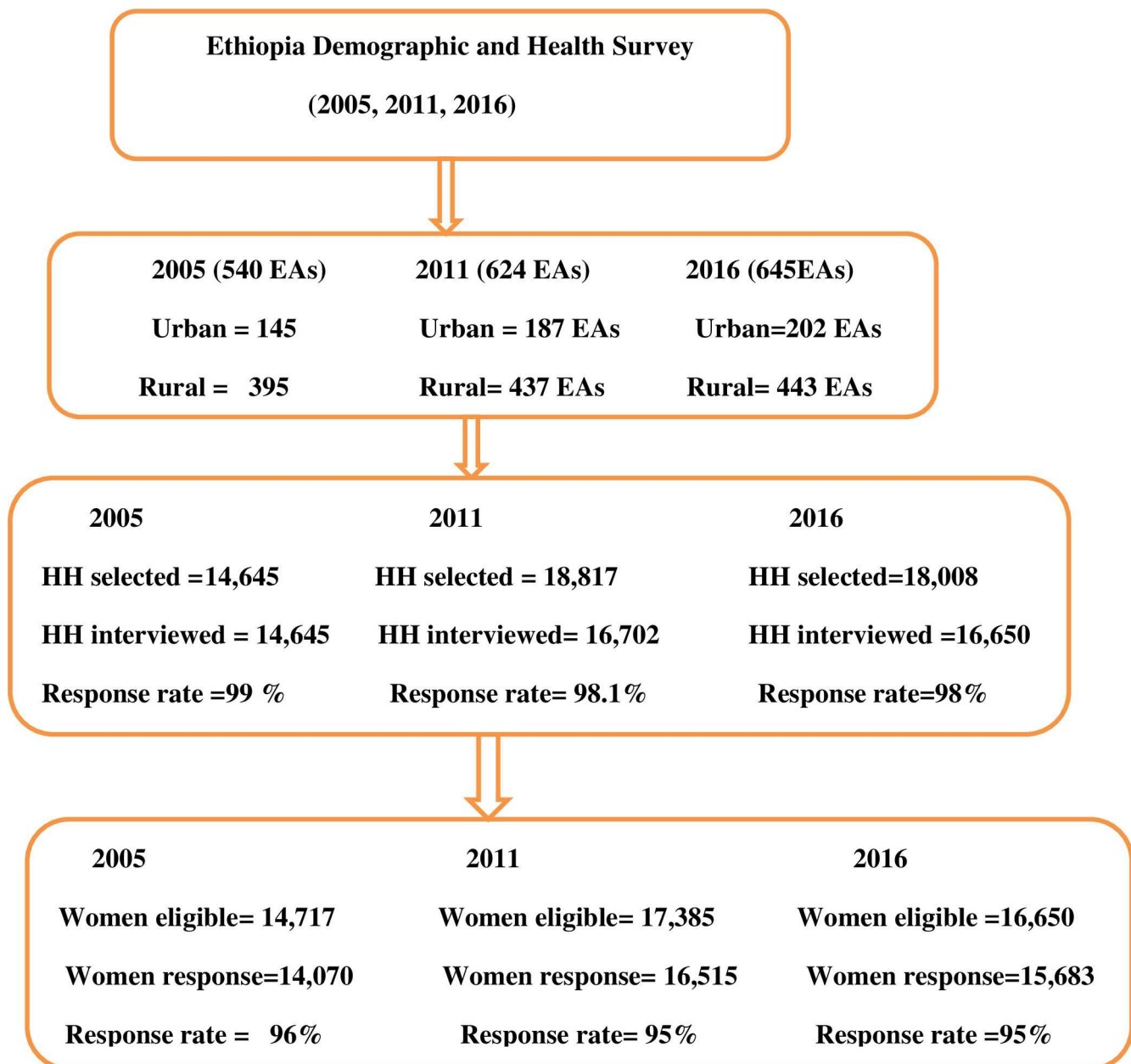

**Fig 1. Schematic presentation of sampling technique of EDHS among women of reproductive age in Ethiopia from 2011–2016.**

**Percentage of anemia by category.** Is obtained by dividing the numerators by the denominator and multiplying by 100.

**Numerators.** Include the classification of the following terms: Any anemia, mild anemia, moderate anemia and sever anemia.

**Any anemia.** Number of non-pregnant women whose hemoglobin count is less than 12.0 grams per deciliter (g/dl).

**Mild anemia.** Number of non-pregnant women whose hemoglobin count is between 11.0 and 11.9 g/dl.

**Moderate anemia.**   Number of non-pregnant women whose hemoglobin count is between 8.0 and 10.9 g/dl.

**Sever anemia.**   Number of non-pregnant women whose hemoglobin count is less than 8.0 g/dl.

## Measurement

**Dependent variable.**   The outcome variable for this study is anemia. In the three consecutive surveys, the level of hemoglobin was measured through adjustments for altitude and smoking, which were already provided in the EDHS data. The hemoglobin level found in the survey data set was already adjusted for altitude using the adjustment formula: adjust = 0.032*alt + 0.022*alt2 and adjHb = Hb—adjust (adjust > 0). The presence of anemia was defined as a categorical variable with pre-defined cut-off points. These include no, mild, moderate, and severe anemia, as recommended by the WHO for women of reproductive age. We have re-categorized anemia again as a binary outcome variable: anemic and non-anemic. We have dichotomized anemia based on prior classifications of the dataset: no anemia, mild anemia, moderate anemia, and severe anemia because of the very small numbers of cases in the categories of severe and mild anemia.

Therefore, women whose hemoglobin level was 120 g/L were considered "anemic," and women whose hemoglobin level was 120 g/L or more were classified as "non-anemic." We have coded "anemia" as "1," otherwise "0," in this study. The independent variables were grouped under the following categories: socio-demographic characteristics, age, educational status (including husband's educational status), religion, ethnicity, marital status, wealth index, and household size. Furthermore, the reproductive variables include age at first pregnancy, number of births, number of antenatal visits, and pregnancy status currently. In addition, factors related to the source of information include the following variables: frequency of listening to the radio, frequency of reading newspapers, and frequency of watching television. Principal component analysis was done to construct the wealth index [37]. The wealth index was created in three consecutive steps. First, the subsets of urban and rural household indicators were built separately. Second, scores were created separately for each household, both in urban and rural areas. Finally, a national-wide wealth index was created by combining both urban and rural settings [38].

## Data collection

Blood specimens were collected for anemia testing from eligible women who voluntarily consented to the testing. Hemoglobin analysis was carried out onsite using a battery-operated portable HemoCue analyzer. Results were given verbally and in writing. Likewise, non-pregnant women were referred for follow-up care if their hemoglobin level was below 7 g/dl. All households in which anaemia testing was conducted received a brochure explaining the causes and prevention of anaemia. At the time of creation of a recode file, an adjustment of the hemoglobin count is made for altitude. Rather than change the cutoff points, the effective hemoglobin count is lowered as altitude increases since oxygen is less available. The adjustment is made with the following formulas: Adjust adjHb if adjusted > 0, where adjust is the amount of the adjustment, alt is altitude in 1,000 feet (converted from meters by dividing by 1,000 and multiplying by 3.3), adjHb is the adjusted hemoglobin level, and Hb is the measured hemoglobin level in grams per deciliter. No adjustment is made for altitudes below 1,000 meters. Both the adjusted and unadjusted hemoglobin counts are included in the recode files.

## Data process and analysis

Initially, the data was checked for the presence of missing data, outliers, and consistency. The complex sampling procedure was also considered by using the SVY STATA command to

control the clustering effect of complex sampling. The data was analyzed using Stata version 14.0 after the two datasets were merged using the Stata command "append." Then, we have created dummy variables for time. Finally, multivariate decomposition analysis was performed to determine the change in anemia and the factors that contributed to the change. The goal of the decomposition analysis was to determine the source of change in anemia among women of reproductive age from 2005 to 2016. The analysis used the output of the logistic regression model to decompose the observed difference in anemia into two components. The first variation was caused by differences in population structure or composition across the survey. The second change was due to a change in the behavior of the survey population, as the change in outcome was due to either a change in population composition, a change in behavior of the population, or both. The observed differences in anemia levels between different surveys were decomposed into characteristics (natural endowment or population composition) and a coefficient (effect of characteristics or behavioral effect). The logit-based difference can be decomposed as [37]

$$Y = F(e^{x\beta}/1 + e^{x\beta}) + \epsilon \qquad (1)$$

$$Ya - Yb = F(e^{xa\beta a}/1 + e^{xa\beta a}) - F(e^{xa\beta a}/1 + e^{xb\beta b}) + \epsilon \qquad (2)$$

$$\Delta Y = [F(e^{xa\beta a}/1 + e^{xa\beta a}) - F(e^{xb\beta a}/1 + e^{xb\beta a})] + [F(e^{xb\beta a}/1 + e^{xb\beta a}) - F(e^{xb\beta b}/1 + e^{xb\beta b}) + \epsilon. \qquad (3)$$

where Y is the dependent variable, X is the independent variable, is the coefficient, and F is the difference between X(ex/1+ ex) and Y. Hence, the result focused on how anemia responded to population composition and behavior and how these factors shaped it across different surveys at different times. The level of statistical significance was set at a P-value of less than 0.05. Analysis was carried out using three consecutive EDHS from 2005 to 2016. These data were appended for analysis of trends in the proportion of anemia. The analyses were done using complex sample analysis to adjust for the cluster sampling design used in the DHS. Whenever there is a non-proportional allocation of samples, the use of sample weights is an important step during analysis. Frequencies were first determined, followed by cross-tabulations to compare frequencies. At the bivariate level, the Pearson chi square ($X^2$) test was used to test the association. Then, bivariate and multivariable logistic regressions were used to identify independent predictors of anemia in the three concurrent EDHS datasets. Results were presented in the form of odds ratios and 95 percent confidence intervals. Statistical significance was set at a p-value of 0.25 and a p-value of 0.05 for binary and multiple logistic regression analyses, respectively. Variables that did not have a significant regression coefficient were removed from the model. Variables that were not significant in the univariate analysis were added back to the model, and their significance was assessed in the presence of other significant variables. Subsequently, the goodness of fit of our final model was tested using the Hosmer-Lemeshow test. Data management and statistical analysis were done with SPSS version 22. In addition, multicollinearity was tested using the variance inflation factor (VIF), and we have a VIF of less than five for each independent variable with a mean VIF of 1.89, indicating there was no significant multicollinearity between independent variables [38].

## Multivariate decomposition analysis of anemia

The decomposition analysis model has taken into account the differences in the characteristics (compositional factors) and the differences due to the effects of characteristics. We have selected the panel regression model from the one-way error component model as it recognizes the distinct heterogeneity in the data. In panel data regression, cross-sectional units or groups

are not the same (they are heterogeneous). Hence, time dummies were created for the 2011 and 2016 surveys, making 2005 a constant that did not need a dummy in this study. As a result, the intercepts of this model vary and are expressed as: Yit = + 1X 1it + 2 X2it + t + Vit. After controlling the compositional factors, 64 points and seven percent (64.7%) of the change in proportion of anemia was due to the difference in effects of characteristics. Among the compositional characteristics, region is the key variable. As a result, women in Afar, Somalia, Harar, and Dire-Dawa had lower anemia rates than women in Tigray. On the contrary, women in Amhara were increasingly affected by anemia in the decade before 2016 compared to women in Tigray. Among the behavioral characteristics, women who had irregular menses, abortions, and the habit of seeking information from various sources (newspapers, television, and radio) had better odds of reducing anemia than their counterparts.

## Poolablity test

Before assessing the validity of the fixed effect method, we need to apply tests to check whether fixed effects (different intercepts in each survey) should indeed be included in the model. Hence, we have done the standard F-test to check the fixed effect against the simple common intercept OLS method. Hence, we have a p-value less than 0.05, so we reject the null hypothesis of common intercepts for all surveys. All year dummies are significant except 2005 (constant). Then, we used the LSDV (fixed effect) model. Only 36.8% of the overall change was due to the difference in compositional characteristics.

## Parameter estimation

Stata 14 was used to fit all the above models to the data set. The variables of the study were carefully selected through an adequate literature review. The F-test was conducted to check the fixed effect against the common intercept model. Hence, we used the LSDV (least square dummy variable) estimator in this study.

## Model selection

The process of model selection was done in two distinct steps. First, we verified that the samples were drawn at random from the population. Second, we have evaluated both the fixed and random effect models. Then, they were compared using the Durbin-Wu-Hausman test. Huasman test is the test of random effect model of in the null hypothesis. We have selected a fixed effect model since the null hypothesis was rejected using a p-value less than 0.05.

## Test of model fit

We have applied the F-test and obtained a significant value, so, we reject the null hypothesis of a common intercept. Moreover, we rejected the null hypothesis in the hausman test, which states that the "difference in the null hypothesis is not systematic " the p-value is less than 0.05. Moreover, we have also checked using the Breusch-pagan LM test and the chow test. All are consistent in showing that the fixed effect model is the right model.

## Ethical approval and consent to participate

Ethical clearance was granted by the federal democratic republic of Ethiopia's ministry of science and technology and the Institutional Review Board (IRB) of ICF. The project numbers were 31406.00.002.12 at a date of September 30, 2008 for the 2005 EDHS(, 31561.00.042.00 at a date of February 28, 2011 for the 2011 EDHS, and FWA-00000845 at a date of June 17, 2017

for the 2016 EDHS. We registered, and we requested datasets from the DHS online archive for this study. Then, we received permission to access and download the data files [39].

## Result

### Descriptive analysis of socio-demographic characteristics

The participants in the study were 28 years old on average, with a standard deviation of 9.6 years. Table 1 presents the socio-demographic characteristics of the study participants. It is apparent from this table that the proportion of educated women showed an increasing pattern. For example, nearly two-thirds (66%) of the study participants were not formally educated in 2005. Further, half (50.2%) and 44.8% of the study subjects had no formal education in 2011 and 2016, respectively. Most (64.9%) of the study participants had no occupation in 2005. On the contrary, less than half (42.3%) of the study participants were jobless in 2011. Most of the study subjects (48.1%) were Orthodox Christians. However, only 2.1% of the study subjects were traditional believers in 2005. On the contrary, we can see that the proportion of Catholics in the study subjects was lowest in 2011 and 2016. There is a clear trend toward increasing wealth quantiles in all three surveys in this study.

### Reproductive health characteristics of the study participants

Table 2 of this study subjects showed that most of the study participants (82.8%) had not read newspapers at all in all surveys. Only few(1.2%) study participants listened at least once per week. Most of the study participants (60.1%) did not listen to radio. Seven point nine percent (7.9%) of study participants had a termination of pregnancy in 2016. On the contrary, only 0.6% of the study participants had abortions. Women who owned mobile phones accounted for 9.8% in 2011. Nearly half (48.9%) of the study participants had menstruation in the past six months in 2005. However, only 44.4% of the study participants had menstruation in the past 6 months in 2016. Only one fifth of the households had electric in 2005. However, one third of the study participants lived in the household with out electricity in 2016.

### Prevalence of anemia among women from 2005–2016

Based on the report of Table 3, Prevalence of anemia among women of reproductive age was 68%, 20.3%, and 27.3% in 2005, 2011 and 2016, respectively.

### Time specific trend of anemia among women from 2005–2016

Fig 2 depicts the trend of anemia; anemia declined in 2011 by 97.3 percentage points relative to the 2005 value. This is because the actual value can be calculated as $(\beta^{0.023}-1)*100$. Furthermore, the prevalence of anemia in 2016 was reduced by 90.8 percentage points relative to the 2011 value. Finally, the prevalence of anemia was reduced by 64.7 percentage points relative to its 2005 value (**Fig 2**).

### Level of anemia among women of reproductive age from 2005–2016

According to Fig 3 report, anemia was classified and displayed based on severe, moderate, and mild anemia, which accounted for 0.6%, 3.4%, and 7.6%, respectively, in 2005. In 2011, the proportions of severe (0.6%), moderate (2.8%), and mild (12.5%) anemia were also recorded. Finally, the prevalence of anemia among women of reproductive age was measured as severe (0.7%), moderate (4.8%), and mild (12%) in Ethiopia in 2016 (Fig 3). Anemia among women of reproductive age varied among regions in Ethiopia in the three DHS surveys. For example, anemia was most prevalent in Oromia (35.3%) and least prevalent in Gambela in 2005.

**Table 1. Socio-demographic characteristics of women from EDHS 2005-2016(N = 46,268).**

| Variable | EDHS-2005 Wt (%) (N = 14,070) | EDHS-2011 Wt (%) (N = 16,515) | p-value | EDHS-2016 Wt (%) (N = 15,683) | p-value | Absolute Difference (% in 2016 - % in 2005) |
|---|---|---|---|---|---|---|
| **Age** | | | | | | |
| 15–19 years | 23.2 | 23.22 | 0.003 | 22.3 | 0.138 | 0.9 |
| 20–29 years | 36.0 | 37.58 | | 36.65 | | 0.65 |
| 30–34 years | 12.8 | 17.66 | | 14.29 | | 1.49 |
| 35–49 years | 28.0 | 21.54 | | 26.76 | | 1.24 |
| **Education** | | | | | | |
| Non educated | 65.9 | 50.12 | 0.000 | 44.84 | 0.000 | 21.06 |
| educated | 34.10 | 49.88 | | 55.16 | | 21.06 |
| **Working status** | | | | | | |
| Work | 34.9 | 57.7 | 0.067 | 46.02 | 0.000 | 21.06 |
| No work | 65.9 | 42.3 | | 53.08 | | 11.12 |
| **Residence** | | | | | | |
| Urban | 17.8 | 32.27 | 0.017 | 34.10 | 0.356 | 16.3 |
| Rural | 82.2 | 67.73 | | 65.90 | | 16.3 |
| **Region** | | | | | | |
| Tigray | 6.5 | 10.46 | 0.000 | 10.73 | 0.000 | 16.3 |
| Afar | 1 | 7.82 | | 7.19 | | 4.23 |
| Amhara | 24.7 | 12.64 | | 10.96 | | 6.19 |
| Oromia | 35.6 | 12.93 | | 12.06 | | 13.74 |
| Somali | 3.5 | 5.53 | | 8.87 | | 23.54 |
| Benishangul | 0.9 | 7.62 | | 7.18 | | 5.37 |
| SNNPRs | 21.3 | 12.32 | | 11.79 | | 6.28 |
| Gambela | 0.3 | 6.84 | | 6.60 | | 9.51 |
| Harari | 0.3 | 6.67 | | 5.78 | | 6.3 |
| Addis Ababa | 5.4 | 10.54 | | 11.63 | | 5.48 |
| Dire Dewa | 0.5 | 6.63 | | 7.21 | | 6.23 |
| **Religion** | | | | | | |
| Orthodox | 48.4 | 42.38 | 0.021 | 40.89 | 0.099 | 7.51 |
| catholic | 1.02 | 1.07 | | 0.58 | | 0.44 |
| Protestant | 16.36 | 13.79 | | 17.94 | | 1.58 |
| Muslim | 32.15 | 37.38 | | 39.59 | | 7.44 |
| traditional | 2.07 | 1.39 | | 0.99 | | 1.08 |
| **Wealth index** | | | | | | |
| Poorest | 17.3 | 18.1 | 0.081 | 18.1 | 0.015 | 0.8 |
| Poorer | 18.8 | 18.4 | | 18.4 | | 0.4 |
| Middle | 19.4 | 18.4 | | 18.4 | | 1 |
| Richer | 18.8 | 19.5 | | 19.5 | | 0.7 |
| Richest | 25.7 | 25.7 | | 25.7 | | 0 |
| **Ethnicity** | | | | | | |
| Amhara | 31.5 | 32.5 | 0.000 | 29.8 | 0.000 | 1.7 |
| Oromo | 32.4 | 32.5 | | 34.0 | | 1.6 |
| Somali | 3.0 | 1.9 | | 2.8 | | 0.2 |
| Sidama | 4.0 | 3.6 | | 4.0 | | 0 |
| Others* | 29.1 | 29.5 | | 29.4 | | 0.3 |

Key:

*- Tigre, Kenbata, Hadiya, Silte, Guarage, Gedio, Berta, Shinasha, Afar, Gambela, Gumuz, Hadrie, Agew, p-value.

**Table 2. Reproductive health and source of information the study participants EDHS 2005-2016(N = 46,268).**

| Variable | EDHS-2005 Wt (%) N = 14,070 | EDHS-2011 Wt (%) N = 16,515 | p-value | EDHS-2016 Wt (%) N = 15,683 | p-value | Absolute Difference (% in 2016 - % in 2005) |
|---|---|---|---|---|---|---|
| **Freq. of reading newspaper** | | | | | | |
| Not at all | 84.6 | 80.3 | 0.000 | 83.6 | 0.000 | 1.0 |
| Less than 1/week | 12.7 | 14.8 | | 11.9 | | 1.2 |
| At least once/week | 1.2 | 4.7 | | 4.4 | | 1.2 |
| **Freq. of watching TV** | | | | | | |
| Not at all | 81.5 | 43.0 | 0.000 | 72 | 0.000 | 8.5 |
| Less than 1/week | 10.5 | 34.7 | | 12.1 | | 1.1 |
| At least once/week | 7.7 | 22.3 | | 15.8 | | 8.1 |
| **Freq. of listening radio** | | | | | | |
| Not at all | 57.3 | 56.2 | 0.000 | 66.9 | 0.000 | 9.6 |
| Less than 1/week | 26.7 | 27.8 | | 16.7 | | 10.0 |
| At least once/week | 15.9 | 16.0 | | 16.5 | | 0.6 |
| **Abortion** | | | | | | |
| Yes | 0.9 | 0.6 | 0.144 | 7.9 | 0.46 | 7.0 |
| no | 93.1 | 91.48 | | 92.1 | | 1.0 |
| **Electricity** | | | | | | |
| No | 81.1 | 72.7 | 0.000 | 70.6 | 0.000 | 11.4 |
| yes | 19.9 | 27.3 | | 29.4 | | 1.9 |
| **Telephone** | | | | | | |
| No | 91.3 | 90.2 | 0.024 | 93.1 | 0.000 | 2.1 |
| yes | 6.8 | 9.8 | | 6.9 | | 0.1 |
| **Menses in the last 6 wk** | | | | | | |
| No | 48.9 | 48.2 | 0.063 | 44.4 | 0.103 | 3.5 |
| Yes | 51.1 | 51.8 | | 55.6 | | 4.5 |

Furthermore, Fig 4 indicates that women in the Oromia region accounted for the most significant proportion (13.5%) of anemia; however, women in Harari accounted for the lowest (0.1%) prevalence of anemia in 2011. In addition, women in the Oromia region (18.4%) were more frail than women in other regions. On the contrary, women in Gambela were the lowest victims of anemia (0.1%) in 2016 (Fig 4).

## Decompositional change of anemia

According to Table 4 description, The decomposition analysis model has taken into account the differences in the features (compositional factors) and the differences due to the effect of characteristics. Only 29.2% of the overall anemia change was due to differences in

**Table 3. The time predictor of anemia among study subjects from 2005-2016(N = 46,268).**

| Year | COR | AOR | P-value |
|---|---|---|---|
| EDHS-2005 | Reference | Reference | |
| EDHS-2011 | 0.023(0.014,032)* | 0.01(0.093,0 .11)** | 0.000 |
| EDHS-2011 | Reference | Reference | |
| EDHS-2016 | 0.076(0.067,0.085)* | 0.034(0.023,0.045)** | 0.000 |
| EDHS-2005 | Reference | Reference | |
| EDHS-2016 | 0.076(0.067,0.085) | 0.13(0.12,0.14)** | 0.000 |

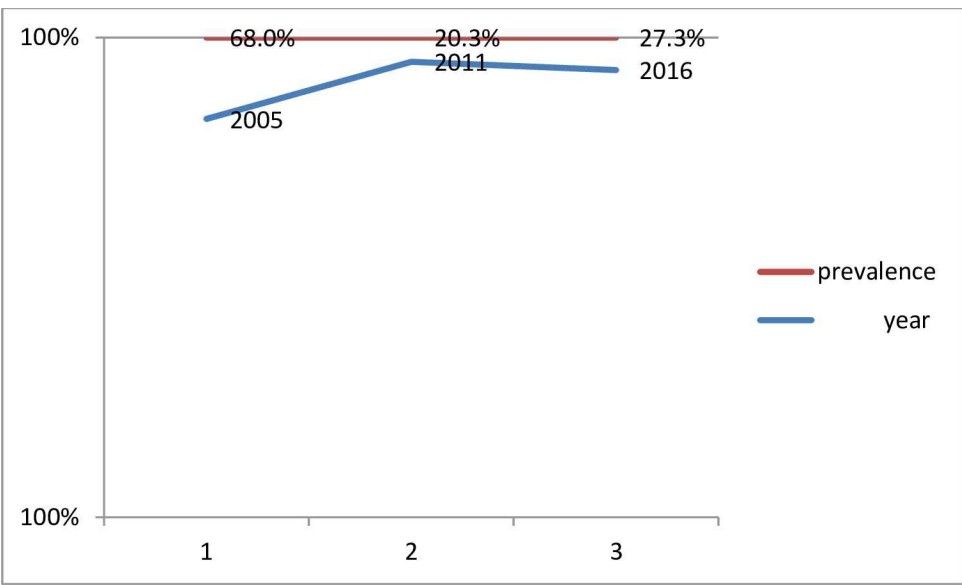

**Fig 2. Trend of anemia among reproductive age group in Ethiopia from 2005–2016.**

characteristics. Among the compositional factors, a very significant contribution to the change in anemia among women of reproductive age was due to the region in which they lived. After controlling the effects of compositional factors, 70.8% of the change in anemia level was due to differences in the effects of characteristics.

## Factors associated with trend of anemia among women from 2005–2016

According to Tables 5 and 6 report,After adjusting for cofounders, region, telephone ownership, menses in the last 6 months, and frequency of reading newspapers were retained as explanatory variables in the final model in 2005. Women in the Afar region were 1.5 times

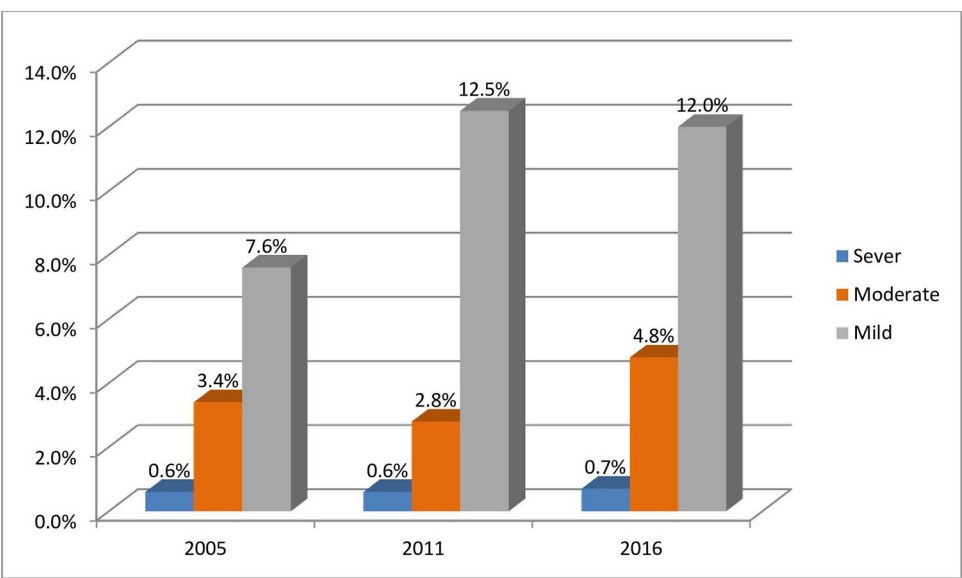

**Fig 3. Severity of anaemia among women of reproductive age group from 2005–2016.**

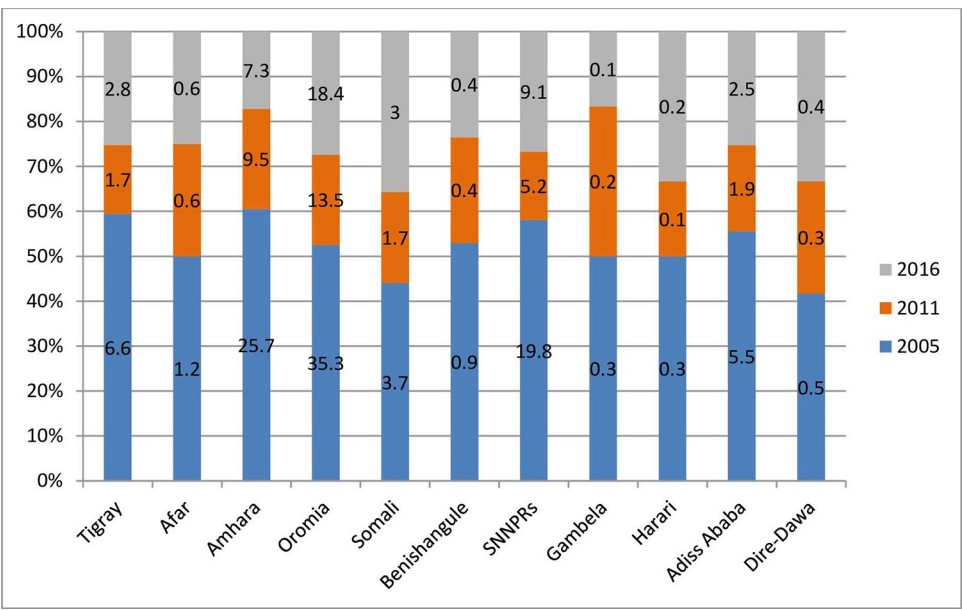

**Fig 4. Regional difference in anaemia among reproductive age group from 2005–2016.**

more likely to suffer from anemia than women in Tigray (AOR = 1.5, 95% CI, 1.1, 2.3).Like-wise, women who live in the Somali region had 1.6 times more odds of getting anemia than women in Tigray (AOR = 1.6, 95% CI, 1.2, 2.1). Women in Ethiopia's southern region were 1.5 times more likely to suffer from anemia than women in Tigray (AOR = 1.5, 95% CI, 1.1, 2). Women had a 21% lower chance of reading newspapers less than once than men (AOR = 0.81; 95% CI: 0.73, 0.9).Women in Afar (AOR = 2.2, 95% CI = 1.5, 3.3), Amhara (AOR = 1.4, 95% CI = 1.2, 1.7), Oromia (AOR = 1.3, 95% CI = 1.04, 1.5), and Somalia (AOR = 3.9, 95% CI = 2.9, 5.3) were more susceptible to anemia than women in Tigray.Similarly, women who lived in Harari (AOR = 1.9: 1.1, 3.8), Adiss Ababa (AOR = 1.8, 95% CI: 1.4, 2.3), and Dire-Dawa (AOR = 2.7, 95% CI: 1.6, 4.7) were more likely to be affected by anemia than women in Tigray. Women who had menstruation in the past six months were 10% less likely to be anemic than their counterparts (AOR = 0.9, 95% CI, 0.85, 0.98). However, women who menstruated in the past 6 months were 20% less likely to be anemic than their counterparts in 2016 (AOR = 0.8, 95% CI, 0.8, 0.9).

In 2011, Catholic women were 2.9 times more likely to be anemic than Orthodox Christian women (AOR = 2, 95% CI: 1.4, 2.9). In addition, Muslim women were 1.5 times more likely to get anemia than Orthodox Christian women (AOR = 1.5, 95%CI, 1.4, 1.7). Women who watched TV fewer than once per week had 10% lower odds than women who did not watch at all (AOR = 0.9, 95% CI, 0.7, 0.9). Furthermore, women who watched TV at least once a week were 20% less likely to have anemia than women who did not (AOR = 0.8, 95% CI, 0.7, 0.98). Women who had previous abortions were 1.4 times more likely to be affected by anemia than their counterparts (AOR = 1.4, 95% CI: 1.2, 1.6). In 2016, Muslims were 1.2 times more likely to be anemic than orthodox Christian women (AOR = 1.2, 95% CI: 1.1, 1.3). Moreover, women who were traditional believers were 2.2 times more likely to get anemia than Orthodox Christian women (AOR = 2.2, 95% CI, 1.6, 2.9). In 2016, women in the richest wealth quantile were 20% less likely to develop anemia than the poorest women (AOR = 0.8, 95% CI: 0.7, 0.9). Compared with the poorest women, richer women were 30% less likely to get anemia (AOR = 0.7, 95% CI, 0.6, 0.8). Women in the middle, particularly wealthy women, were 30%

**Table 4. Decomposition change of anemia among women in reproductive age (N = 46,256).**

| Category | Difference due to characteristics (E) | | | Difference due to coefficients(C) | | |
|---|---|---|---|---|---|---|
| | coefficient | percent | p-value | coefficient | percent | p-value |
| **Educational status** | | | | | | |
| no formal education | Reference | | | | | |
| formal educated | -0.00739 . | -7.39 | 0.121 | -0.0614184 | -61.4 | 0.000 |
| **Religion** | | | | | | |
| Orthodox | Reference | | | | | |
| Catholic | 0.0527 | 52.7 | 0.019 | 0.0240821 | 24.0 | 0.133 |
| Protestant | 0.0386 | 38.6 | 0.000 | 0.0079874 | 7.9 | 0.139 |
| Muslim | 0.0455 | 45.5 | 0.000 | 0.0065698 | 6.5 | 0.001 |
| Traditional | 0.0365 | 36.5 | 0.043 | 0.0192629 | 19.2 | 0.004 |
| **Region** | | | | | | |
| Tigray | Reference | | | | | |
| Afar | 0.01485929 | 14.8 | 0.000 | 0.0145 | 14.6 | 0.000 |
| Amhara | 0.0103734 | 10.3 | 0.242 | 0.0194898 | 19.4 | 0.039 |
| Oromia | 0.0130753 | 13.0 | 0.166 | 0.0513826 | 51.1 | 0.000 |
| Somali | 0.0241835 | 24.1 | 0.000 | 0.0232967 | 23.2 | 0.000 |
| Benishangul | -0.001869 | -1.86 | 0.862 | -0.011323 | -11.3 | 0.325 |
| SNNPR | -0.0407727 . | -40.7 | 0.000 | -0.000865 | -0.8 | 0.937 |
| Gambela | 0.0332232 | 33.2 | 0.0008 | -0.009723 | -9.7 | 0.574 |
| Harari | 0.01339532 | 13.3 | 0.0999 | 0.0101653 | 10.1 | 0.000 |
| Addis Ababa | 0.0484661 | 48.4 | 0.0209 | 0.049897 | 49.8 | 0.000 |
| Dire-Dawa | 0.01553923 | 15.5 | 0.1237 | 0.01275 | 12.7 | 0.000 |
| **Residence** | | | | | | |
| Urban | Reference | | | | | |
| Rural | 0.014208 | 14.2 | 0.110 | 0.0262 | 26.2 | 0.006 |
| **Marital status** | | | | | | |
| never married | Reference | | | | | |
| Married | 0.0159146 | 15.9 | 0.004 | .0095065 | 9.5 | 0.107 |
| live together | 0.0204363 | 20.4 | 0.142 | -0.0375 | -37.5 | 0.011 |
| widowed | 0.0269161 | 26.9 | 0.024 | 0.0355 | 35.5 | 0.005 |
| divorced | 0.0094399 | 9.4 | 0.347 | -0.0072 | -7.2 | 0.501 |
| not live together | 0.0270724 | 27 | 0.070 | 0.0257 | 25.7 | 0.106 |
| **Wealth index** | | | | | | |
| poorest | Reference | | | | | |
| poorer | -0.0154 | -15.4 | 0.036 | -0.0019 | -1.9 | 0.805 |
| middle | -0.0186 | -18.6 | 0.014 | -0.00095 | -0.9 | 0.906 |
| Richer | -0.0308 | -30.8 | 0.000 | -0.0167776 | -16.8 | 0.039 |
| richest | -0.01876 | -18.7 | 0.053 | 0.0550452 | 55 | 0.000 |
| **Abortion** | | | | | | |
| no | Reference | | | | | |
| yes | -0.00536 | -5.3 | 0.822 | -0.0273 | -27.3 | 0.001 |
| **Menstruate in the last 6 wk.** | | | | | | |
| no | Reference | | | | | |
| yes | -0.0017 | -1.7 | 0.238 | -0.0068 | -6.8 | 0.159 |
| **Freq.of reading newspaper** | | | | | | |
| Not at all | Reference | | | | | |
| Less than once/week | 0.0033 | 3.3 | 0.000 | -0.0122 | -12.2 | 0.099 |

(*Continued*)

**Table 4.** (Continued)

| Category | Difference due to characteristics (E) | | | Difference due to coefficients(C) | | |
|---|---|---|---|---|---|---|
| | coefficient | percent | p-value | coefficient | percent | p-value |
| At least once/week | -0.02867 | -28.7 | 0.766 | -0.0026 | -2.6 | 0.829 |
| Almost everyday | -0.01235 | -12.3 | 0.326 | 0.03975 | 39.7 | 0.202 |
| **Frequency of watching Tv** | | | | | | |
| Not at all | Reference | | | | | |
| Less than once/week | -0.0050154 | -5.0 | 0.609 | 0.0129586 | 12.9 | 0.029 |
| At least once/week | -0.0072573 | -7.2 | 0.476 | -0.00597 | -5.9 | 0.000 |
| Almost everyday | 0.036423 | 36.4 | 0.544 | 0.02012 | 20.1 | 0.000 |
| **Frequency of listening radio** | | | | | | |
| Not at all | Reference | | | | | |
| Less than once/week | -0.0157 | -15.7 | 0.022 | -0.01042 | -10.4 | 0.000 |
| At least once/week | -0.0348 | -34.8 | 0.000 | -0.01718 | -17.1 | 0.000 |
| Almost everyday | 0.00035 | 0.35 | 0.980 | 0.008913 | 8.9 | 0.000 |

**Table 5. Socio-demographic predictors of anemia among study participants (N = 46,256).**

| Variables | EDHS = 2005 | | EDHS 2011 | | EDHS = 2016 | |
|---|---|---|---|---|---|---|
| | COR(95%CI) | AOR(95%CI) | COR(95%CI) | AOR(95%CI) | COR(95%CI) | AOR(95%CI) |
| **Marital status** | | | | | | |
| married | - | - | Ref. | Ref. | Ref. | Ref |
| Single | - | - | 0.74(0.67,0.80)* | 0.8(0.7,0.9)** | 0.8(0.7,0.85)* | 0.8(0.7,0.9)** |
| Widowed | - | - | 1.60(1.4,1.99)* | 1.7(1.4,2.0)** | 0.79(0.6,0.99)* | 0.7(0.6,0.9)** |
| Divorced | - | - | 0.91(0.76,1.08) | 0.9(0.79,1.1) | 0.7(0.6,0.8)* | 0.8(0.6,0.9)** |
| **Region** | | | | | | |
| Tigray | Ref. | Ref. | Ref. | Ref. | Ref | Ref. |
| Afar | 1.6(1.1,2.4)* | 1.5(1.1,2.3)** | 3.4(2.3,4.9)* | 2.2(1.5,3.3)** | 3.0 (2.1,4.4)* | 2.3(1.6,3.4)** |
| Amhara | 1.1(0.9,1.2) | 1.05(0.9,0,1.2) | 1.5(1.3,1.8)* | 1.4(1.2,1.7)** | 0.74(0.6,0.8)* | 0.7(0.6,0.8)** |
| Oromia | 0.95(0.81,1.10) | 0.91(0.78,1.1) | 1.6(1.3,1.93)* | 1.3(1.04,1.5)** | 1.4(1.2,1.7)* | 1.1(1.5,3.4)** |
| Somali | 1.6(1.2,2.05)* | 1.5(1.1,1.97)** | 5.9(4.5,7.9)* | 3.9(2.9,5.3)** | 5.5(4.4,6.9)* | 4.2(3.2,5.4)** |
| Benishangul | 0.94(0.63,1.41) | 0.90(0.60,1.35) | 1.7(1.2, 2.5) | 1.3(0.8,1.9) | 1.1(0.7,1.7) | 1.0(0.6,1.5) |
| SNNPRs | 0.79(0.6,0.93)* | 0.76(0.65,0.89) | 1.0(0.87,1.3) | 0.8(0.7,1.1) | 1.2(0.9,1.4) | 1.0(0.8,1.2) |
| Gambela | 1.2(0.6,2.3) | 1.13(0.57,2.23) | 1.7(0.92,3.0) | 1.4(0.8,2.6) | 1.4(0.6,2.6) | 1.2(0.6,2.4) |
| Harari | 1.0(0.5,2.0) | 0.97(0.48,1.95) | 2.3(1.2,4.4)* | 1.9(1.01,3.8)** | 2.1(1.1,4.1)* | 2.1(1.1,4.1)** |
| Adis Ababa | 1.0(0.84,1.2) | 0.99(0.79,1.25) | 1.5(1.2,1.9)* | 1.8(1.4,2.3)** | 1.0(0.8,1.3) | 1.3(1.0,1.6) |
| Dire Dewa | 1.21(0.70,2.1) | 1.18(0.68,2.05) | 3.2(1.9,5.4)* | 2.7(1.6,4.7)** | 2.2(1.3,3.4)* | 2.1(1.4,3.4)** |
| **Religion** | | | | | | |
| Orthodox | - | - | Ref . | Ref. | Ref. | Ref. |
| catholic | - | - | 1.51(1.1,2.1)* | 2.0(1.4,2.9)** | 1.08(0.71,1.7) | 0.8(0.6,1.4) |
| Protestant | - | - | 0.98(0.88,1.09) | 1.2(1.0,1.4) | 1.3(1.2,1.46) | 1.1(0.9,1.3) |
| Muslim | - | - | 1.8(1.6, 1.9)* | 1.5(1.4,1.7)** | 1.8(1.6,1.9)* | 1.2(1.1,1.3)** |
| traditional | - | - | 0.8(0.5,1.1) | 0.8(0.6,1.2) | 3.4(2.6,4.4)* | 2.2(1.6,2.9)** |
| **Wealth index** | - | - | | | | |
| Poorest | - | - | Ref. | Ref. | Ref | Ref. |
| Poorer | - | - | 0.94(0.84,1.07) | 0.03(0.9,1.1) | 0.65(0.58,0.7)* | 0.8(0.7,0.9)** |
| Middle | - | - | 0.84(0.74,0.95)** | 0.9(0.8,1.0) | 0.59(0.53,0.7)* | 0.7(0.6,0.8)** |
| Richer | - | - | 0.85(0.75,0.96)** | 0.9(0.8,1.1) | 0.54(0.51,0.6)* | 0.7(0.6,0.8)** |
| Richest | - | - | 0.79(0.70,0.88)** | 1.1(0.9,1.3) | 0.54(0.4,0.6)* | 0.7(0.6,0.8)** |

**Table 6. Source of information and women reproductive health characteristics.**

| | COR(95%CI) | AOR(95%CI) | COR(95%CI) | AOR(95%CI) | COR(95%CI) | AOR(95%CI) |
|---|---|---|---|---|---|---|
| **Reading newspaper** | | | | | | |
| Not at all | Ref. | Ref. | | | Ref. | Ref. |
| < one /week | 0.8(0.7,0.9)* | 0.9(0.87,1.1) ** | - | - | 0.6(0.5,0.7) * | 0.8(0.7,0.9)** |
| One /week | - | - | - | - | 1.1(0.8,1.2) | 0.8(0.9,1.4) |
| **Watching Tv** | | | | | | |
| Not at all | - | - | Ref. | Ref | - | - |
| Less than 1/week | - | - | 0.76(0.7,0.8) * | 0.9(0.7,0.9)** | - | - |
| At least 1/week | - | - | 0.74(0.6,0.8)* | 0.8(0.7,0.98)** | - | - |
| **Freq.lis radio** | | | | | | |
| Not at all | - | - | Ref. | Ref. | Ref. | Ref. |
| Less than 1/week | - | - | 0.87(0.8,0.9) * | 0.9(0.86,1.1) | 0.9(0.8,0.99) | 1.1(0.9,1.2) |
| At least 1/week | - | - | 0.7(0.69,0.8)* | 0.8(0.7,0.9)** | 0.8(0.8,0.9)* | 0.9(0.88,1.1) |
| **Abortion** | | | | | | |
| Yes | - | - | Ref. | Ref. | - | - |
| no | - | - | 1.4(1.2,1.6)* | 1.4(1.2,1.6)** | - | - |
| **Telephone** | | | | | | |
| yes | Ref. | Ref. | - | - | - | - |
| no | 1.2(1.1,1.4)* | 1.4(1.2,1.62)** | - | - | - | - |
| **Menses in 6 wk** | | | | | | |
| yes | Ref. | Ref. | Ref. | Ref. | - | - |
| no | 0.9(0.8,0.96)* | 0.9(0.85,0.98)** | 0.87(0.8,0.9)* | 0.8(0.8,0.9)** | - | - |

**Key:** COR: Crude Odds Ratio, AOR: Adjusted Odds Ratio, CI: Confidence Interval, P-value < 0.25, ** = p < 0.05.

less likely to get anemia than the poorest women (AOR = 0.7, 95% CI, 0.6, 0.8). Finally, poorer women were 30% less likely to get anemia than the poorest women (AOR = 0.7, 95% CI, 0.6, 0.8).

## Discussion

Although many policies were implemented to minimize the magnitude of anemia, it is still high among women of reproductive age in Ethiopia. Further, although many studies have been conducted in the past decades regarding anemia, none of them have reported on the national trend and factors associated with the trend of anemia among women in Ethiopia from 2005 to 2016 [39–42]. Therefore, we have found that the prevalence of anemia among women of reproductive age has varied significantly in opposing directions in the last decade. This might be due to drought, political instability, economic instability, and food insecurity.

In this study, we discovered a trend of an absolute difference in the proportion of anaemia in three consecutive Ethiopian surveys. It decreased significantly by 47.7 percentage points from 2005 to 2011 in Ethiopia. This finding is consistent with findings about the trend of anemia in most African countries [8].

The possible similarities might be due to effective policy implementation in areas such as malaria diagnosis and treatment, national policy on sanitation, iron and folic acid fortification, family planning, insecticides for household use, and iron fortification. Agriculture's involvement has increased income, iron crop production, paultary and dietary diversity, and the health sector has effectively provided iron supplementation, family planning, malaria prevention, and environmental sanitation during health services [43–45]. Furthermore, the

prevalence of anemia increased by seven percentage points (7%) from 2011 to 2016. This finding is in contrast with the finding of a previous study done in Uganda, which showed a 17 percentage point reduction among women of reproductive age from 49% to 32% in 2006–2016, respectively [46]. The possible explanation might be due to income differences and political stability between countries, which have laid the foundation for food security and effective health service delivery.

In this study, we have assessed the time-invariant factors of anemia, which previous studies failed to show using the least squares dummy variable estimator. Generally, women with a lower wealth index were more likely to develop anema than their counterparts. This finding is consistent with the previous study's finding [47].

This might be due to the fact that rich women have a greater chance of getting a balanced diet, a reduced risk of infection (morbidity), and better access to and utilization of health care services [48, 49]. Women in Afar, Oromia, Somalia, Harari, Addis Abeba, and Dire-Dawa had a higher risk of anemia than women in Tigray. This is consistent with other studies conducted elsewhere [16, 50, 51]. The reason might be due to variation attributable to cultural practices, societal beliefs, geographical conditions, climatic conditions, disease burden, access to health care services, and dietary-associated factors between regions [52, 53]. Furthermore, differences in maternal health care utilization [54], food consumption preferences [55] and differences in availability of healthcare facilities [56]. Women in less developed regions or states were more affected by anemia in this study than women in more developed regions or states [57]. The possible explanation could be a lack of clean water and unimproved latrine facilities, which result in soil-transmitted disease [58] and increased chance of anemia [59].

Single women had a lower chance of getting anemia than their counterparts. This finding is consistent with the findings of the study done in Rwanda [60]. The possible explanation might be due to variation in income, food habits, and menstrual regularity. On the contrary, a study found that married women were more likely to develop anemia than their Nigerian counterparts [61]. This could be explained by fetal iron depletion, blood loss during childbirth, and short birth intervals among married women, all of which contribute to short birth intervals. This study revealed that widowed and separated women had higher odds of having anemia than their counterparts. This might be explained by the fact that widows and women who have been separated from their husbands are vulnerable to economic deprivation, hunger, starvation, and a lack of access to health care [48, 62, 63].

Women can reduce their anemia by listening to the radio and reading the news more frequently. This finding is consistent with the previous study conducted in Indonesia [64]. The possible explanation might be associated with the fact that improved knowledge and behavioral change occurred as women got more information on anemia. This study has found that women who had previous abortions had higher odds of getting anemia than their counterparts. This finding is consistent with the study done in West Arsi, the Oromia region, that showed the history of abortion was significantly associated with anemia [65]. This could be related to the fact that anemia is caused by hemorrhage. Women who had a history of heavy menstruation in the last 6 weeks were more likely to be affected by anemia among women of reproductive age in this study [66, 67]. The possible explanation might be associated with prolonged bleeding [68, 69]. Lack of electricity was associated with anemia among women of reproductive age. This finding was consistent with findings from previous studies [70, 71]. The possible expulsion could be due to increased utilization of cooking fuel, which may in turn aggravate anemia [72]. The possibility is that the causal link between biomass smoke and anemia is due to its ability to induce systemic inflammation [73] which can be indicative of carbon monoxide and transitional metal content [74]. The provision of electricity and other clean energy sources may therefore eliminate the need for biomass and reduce the incidence of

anemia [75]. Women who had no mobile phone were more likely to be anemic than their counterparts. This finding is consistent with previous studies [76]. The possible explanation might be associated with information they can get from different social media and the exchange of food items and medical advice through different applications [77].

### Limitation of the study

This study is one of the few studies that report the level, trend, and predictors of anemia among reproductive women in Ethiopia at the national level. As a result, it employs a sufficient sample size, making the data more reliable. Furthermore, standard national tools and methods were used to make the measurements more accurate through an expert data collection process. On the contrary, this study might have recalled some past events. Besides, this study failed to incorporate factors related to the household living condition, such as ITN, source of drinking water, and type of toilet facilities. Hence, the findings of this study should be interpreted in light of these limitations.

### Conclusion and recommendation

This study highlights that the trend of anemia among women of reproductive age has been fluctuating in the past 10 years, from 2005 to 2016. Low socio-demographic status and a lack of basic household facilities are the main factors associated with anemia among women in Ethiopia. Therefore, health policy makers, program designers, and local government authorities should enhance the socio-demographic characteristics and basic infrastructure of the community. Furthermore, they should design strategies for extensive media coverage about anemia. Finally, resources and social services should be distributed among regional states in the most equitable manner to prevent anemia throughout Ethiopia among women of reproductive age, who are an easily vulnerable group due to their biological and social roles differing from other population groups.

### Public health implication

This study contributes input for public health policy to focus on media coverage to prevent anemia and minimize its impact on the general health condition of women. Furthermore, equitable health services and agricultural food production among regions are needed to prevent anemia. Finally, policies and programs should ensure equitable access to education and wealth among women in the country.

### Supporting information

**S1 Data.**
(SAV)

**S2 Data.**
(SAV)

**S3 Data.**
(SAV)

### Acknowledgments

We are grateful to the MEASURE DHS program for permitting us to obtain and use the 2016 EDHS data set.

## Author Contributions

**Conceptualization:** Berhan Tsegaye Negash.

**Data curation:** Berhan Tsegaye Negash.

**Formal analysis:** Berhan Tsegaye Negash.

**Methodology:** Berhan Tsegaye Negash, Mohammed Ayalew.

**Software:** Berhan Tsegaye Negash.

**Validation:** Berhan Tsegaye Negash.

**Writing – original draft:** Berhan Tsegaye Negash, Mohammed Ayalew.

**Writing – review & editing:** Berhan Tsegaye Negash, Mohammed Ayalew.

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
