## [Decision Letter · Decision Letter 0]

19 Apr 2022

PONE-D-21-33298Trend of anemia among women reproductive age group in Ethiopia from 2005-2016: A Further analysis of Ethiopian demographic health survey.PLOS ONE

Dear Dr. Tsegaye,

Thank you for submitting your manuscript to PLOS ONE. After careful consideration, we feel that it has merit but does not fully meet PLOS ONE’s publication criteria as it currently stands. Therefore, we invite you to submit a revised version of the manuscript that addresses the points raised during the review process.

Please could you complete revisions according to those set out by reviewer 1 who has set out changes per line. 

We look forward to receiving your revised manuscript.

Kind regards,

Caroline Anita Lynch

Academic Editor

PLOS ONE

Journal Requirements:

4. We note you have included a table to which you do not refer in the text of your manuscript. Please ensure that you refer to Table 1 and 2 in your text; if accepted, production will need this reference to link the reader to the Table.

Reviewers' comments:

Reviewer's Responses to Questions

**Comments to the Author**

1. Is the manuscript technically sound, and do the data support the conclusions?

Reviewer #1: Partly

Reviewer #2: Yes

2. Has the statistical analysis been performed appropriately and rigorously? 

Reviewer #1: No

Reviewer #2: Yes

3. Have the authors made all data underlying the findings in their manuscript fully available?

Reviewer #1: No

Reviewer #2: Yes

4. Is the manuscript presented in an intelligible fashion and written in standard English?

Reviewer #1: No

Reviewer #2: Yes

5. Review Comments to the Author

Reviewer #1: Thank you very much for the opportunity of reviewing this work. Anemia is a major public health problem globally that affects the most vulnerable, including women of childbearing age, pregnant women, and children under 5-years, mostly in low- and middle-income countries. Analysis of its trends and associated factors are relevant to tackle down anemia through policies and programs focused on its various determinants.

General comment: Authors wrote very poor English, many of sentences were hard to understand. I strongly suggest the authors search for English proofreading services / checking by English native speakers to enhance English writing and reading. Data analysis was very poor which need to re-check, including the selected variables.

Title: This manuscript reported both the trend and factors associated with anemia among women of reproductive age. I suggested to revise the title of your manuscript to include factor associated with anemia. Also, women of reproductive age means the group of women aged 15-49 years itself, so, it is better to remove word ‘group’ after women of reproductive age throughout this manuscript. In addition, in my understanding we usually call: Demographic “and” Health Survey (DSH), Please revise the correct word of this survey? if my comment is correct, you need to revise this word throughout manuscript.

Abstract

Background. Please check the information regarding to anemia is the most common cause of indirect morbidity?

Methods. Please revise these paragraph, many information is not related to the result section, particularly the odds ratio, 95% confidence interval are not showing in the result section at all.

Result. Please revise all sentences because in the main text did not show the results of the positive or negative association of several factors with anemia.

Conclusion. Authors recommended several unrelated ideas which is not consistent with the findings from this study. Authors should revise this paragraph.

The introduction or background section: It is better to reconstruct this paragraph by describing the situation of anemia from global, regional (high-income/low-income countries), and country levels, respectively. Additionally, this study aimed to analyze the trend of anemia in Ethiopia, however, in this section is not describe about the trend of anemia as the priority problem in this study. Moreover, this section is significantly long text, I strongly recommend the authors divided this section at least two paragraphs.

Line 63. It is better to give the definition of anemia for a common understanding for the readers.

Line 66. Please revise this sentence with evidence regarding the cause of anemia, because “it might be caused” is not a clear statement because anemia can caused by infections, genetic conditions,… etc. So, Please clarify the cause of anemia.

Line 74-76. Can you give more information how this sentence related to anemia among women of reproductive age, particularly what kind of extra foods and supplement needed during pregnancy and lactation? Also, what is the mechanism of the physiological changes, metabolism, growth of fetus and breast feeding during pregnancy and lactating affect to anemia among these women?

Line 80-81. Physiological and nutritional change need to clarify for the completeness of sentence and can be merged with line number 74-76.

Line 84. What is the meaning of grand multiparity?

Line 90-92. In my opinion, using the word “previous studies” instead of the word “former studies” might be better, which is commonly used in scientific report. In addition, I am not sure about the intention of author would like to talk about this sentence on how is related to anemia research?

Line 94-95. What kind of the existing policies and programs which is addressed to anemia problem among women of reproductive age in Ethiopia? What is the barriers of these policies and programs which could not reduce the prevalence of anemia among these women in Ethiopia? Author might additional raise these important problem in this manuscript.

Line 99-101. Please revise this sentence: First point: This study “aimed” to analyze the trend and factors associated with anemia among…., Second point: If you use the word women of reproductive age, you need to use the same wording throughout manuscript to avoid the misunderstanding. Third point: the national data using to analyze in this study should be clearly with exact name and years.

Methods

General comment on study setting section. This paragraph is quite complicated in term of sentence order and unrelated ideas. This paragraph should be clearly provided information regarding to settings and locations where the data were collected and locations in which the study was carried out, including the country, region, city. Also, authors could provide other information about settings and locations that could have affected a study’s external validity such as social, economic, cultural environment and other special aspect of study settings, if applicable. In addition, I would like to suggest the authors to provide the decision trees that would be innovative to explore such data (merely a thought) in order to show the readers on how the sample were selected and excluded at each stages.

Line 107-109. Please revise this sentence started from About 20%..... until in rural areas by giving the exact numbers/percentages of people living in rural and urban areas. Nearly 80% is not exact number of 80%.

Line number 109-110. 1st point, I am not sure that we can say 4.6 individuals/persons or what is the meaning of 4.6 persons? 2nd point, This study used three different dataset of 2005, 2011 and 2016, but in this sentence mentioned only the survey conducted in 2016.

Line 111-137. Data source/study design, study population and sampling technique. The text of this section can be divided in three paragraphs separately under this sub-heading.

- Data source should be clearly mentioned the source of datasets using/analyzing in this study, year of conducting these surveys, type of data (primary/secondary datasets?) along with providing information about these surveys.

- Study population.

- Sampling technique. In my understanding the DHS survey used the multi-stage cluster sampling technique. Authors can re-check in the report of each original surveys, and please provide the related information into this section.

- DHS surveys typically provide altitude and smoking-adjusted levels of hemoglobin. In your study, I truly believe that smoking is not a concern; however, altitude might be. During data collection and building databases, some factors were used to adjust the levels of hemoglobin?

Line 116. Authors should provide the full text for EDHS as the first time state follow by the abbreviation in blanket. Also, this study used three cross-sectional data of 2005, 2011, and 2016 but the sampling frame mentioned only in 2016. Can authors provide the reason on this matter for clarification.

Line 165. non-governmental organizations.

Line 126. What is the different between cluster and EA in this sentence?

Line 127-134. 17,067 households had women of reproductive age. Can authors clarify why all of these women were not include? What is the eligible selection criteria?

Study variables

General comment for this section. This section is absolutely complicated with unnecessary information. It is better to separate this section into two paragraphs to show the dependent and independent variables separately, and under the independent variables can be divided in two categorization of socio-demographic characteristics of the participants follow by reproductive characteristics. Furthermore, due to this section describe specifically for the study variables, therefore, the information regarding to data collection and its process should be mention in above section of the data source/study design, study population and sampling technique.

Many independent variables were not mentioned in this section, but showed in result’s tables such as work status, residence, region, frequency of reading new papers, frequency of watching TV, Abortion, electricity, telephone, menstruated in the last six months. On the other hands, many independent variables mentioned in this section, but not showed in result’s tables such as husband educational status, age at first pregnancy, number of birth, number of antenatal visit, and current pregnancy. Authors must re-organize it clearly, and each variables must show its categorization, for example: education (uneducated vs educated).

Line 139-144. Authors provide two times regarding the definition of anemia among non-pregnant women, authors can select one of them. Also, please provide full definition regarding of anemia among pregnant and non-pregnant women because its definitions used the different cut point for mild, moderate and severe anemia.

Line 180. Please give the full text of CSA before it first statement of its abbreviation.

Data management and statistical analysis.

In statistical section, please refer to any post-hoc corrections to correct for multiple comparisons during your statistical analyses. If these were not performed please justify the reasons. Additionally, in your statistical analyses, please state whether you accounted for clustering by Kebeles or region?. For example, did you consider using multilevel models?

Line 187 and 201-201. Which version of SPSS used to analyze the data? And it should follow by showing the name of its company?

Line 188. How many datasets were analyzed for this study? Three or four? And it should be mentioned the year of each datasets!!

Line 189-190. Authors must understand this section which must provide only the statistical tests used/analyzed by yourselves.

Line 191. Are authors used the sampling weight for all analyzes?

Line 193-194. Which statistical tests whether chi-square or logistic used to analyzed the date?

Line 194: In which criteria that authors selected variables into multivariable logistic regression? And how do you treat with multicollinearity problem? These can be strengthened with a reference at least.

Line 195-196. I did not see the odds ratio and 95% confidence interval in the result’s tables.

Line 196-197. Which number of p-value consider as statistical significant 0.25 or 0.05 and please cited a reference?

Line 198-200. Please rephrase this sentence, I don't understand. Also, what is the different between univariate and bivariate analyses?

Line 203-208. Ethical consideration: Understandable, the secondary data analysis is not required the double ethical approval, however, authors should provide additional details regarding participant consent as well as which organization/agencies provided ethical approval for these surveys. Please ensure that you have specified what type you obtained (for instance, written or verbal, and if verbal, how it was documented and witnessed). If the need for consent was waived by the ethics committee, please include this information.

Result

Can authors show the number of participant in result’s tables 1 and 2 for each categories of each variables follow by its percentages?

Many independent variables were not showed in result’s tables 1 and 2 but mentioned in method section such as husband education, marital status, age at first pregnancy, number of birth, number of antenatal visit, current pregnancy. Authors need to re-check for clarification!!!

Authors did not show the result of logistic regression analyses with the result of odds ratio and 95% confidence interval?

Why the menstruated information was asked in the last six weeks? Because the menstruation were usually happened as monthly basis.

Line 210. Please revise this sub-heading. The correct word is “characteristics” and please re-check this word in other section as well.

Line 211. Mean age of the participants was 28, is ±9.6 years the standard deviation? Please provide clearly in such phrase.

Line 211-214. Majority mean more than half. Please revise the related sentences, along with providing full description of result of selected variables from each year of surveys 2005, 2011, and 2016, respectively. Due to this study used three different datasets from different years, authors should be careful to provide the clear description.

Line 216-217. In result section, authors should provide the exact number of percentage of each selected variable. The least proportion is not a common used in scientific report.

Table 1. (i) Please revise the name of this table, (ii) showing two p-value in a table can cause misunderstanding, other might select one of them and need to explain in statistical analyses section. (iii) age category is not in the similar frequency, for example: 15-19 years is within 5 years but other category of 20-29 years is within 10 years, do you have any reference to strengthen this categorization?

Line 226-230. I recommend to re-interpret the results of the table 2?? A swell as revising the name of the table 2.

Line 237-249. It is better to show the figures before or after the result’s interpretation. Authors should interpreted each figure with a separate texts.

Line 239-240. What is the different between 47.7% and 47.7 percentage points?

Line 250-256. I am not sure that the result in this section came from which tables? The results are not show!!!!

Discussions:

In your discussions, please take care to avoid statements implying causality from correlational research. For example, avoid the use of terms such as "increased/reduced risk", "influenced by", “likely to” or “resulted in." Instead, consistently use terms such as "associated with" or "associations."

Line 261. Anemia vs anaemia are American and British English, please select one of them for whole manuscript.

Line 261-265. Authors don’t need to give the definition of anemia in this section which is already mentioned in the background section.

Line 266. In statistic, to show the full number like 68% must follow by dots zero as 68.0%, authors should revise other full numbers as well.

Line 265-277. It is good to compare the prevalence with other countries, however, the author should try to find the evidence to answer why the trend of prevalence of anemia was significantly reduced from 2005 to 2011, but increased from 2011 to 2016 in Ethiopia? This should be the main discussion regarding to the findings from this study. Had it any policy/intervention or event which influenced to this unstable trends? The comparison of prevalence of anemia among children and pregnant women with women of reproductive age might led to unrelated ideas, because the study participants are different. Also, authors should provide the year of the references’ cited for example in line 275-277, recent national DHS and other surveys were conducted in which years? is that only one reference for respective countries?

Line 277-278. What is the reduction of anemia among women of reproductive age important for child health?

Line 228-280. What is the meaning of this sentence “anaemia prevalence remains high and haemoglobin levels remain low in the low income countries”?.

Line 280-281. “If these trends are continued, the likelihood of reducing anemia by half from 2011 levels by 2025 in all regions among reproductive age women”. This prediction might be happened and might not be happened as well, the scientific research should come up with the fact and real evidence.

Line 282-283. What is the group of lower wealth indexes? According to tables’ result showed five categories from poorest to richest group? Which group is the reference group?.

Line 283-285. Can authors provide more evidence regarding to women from high socioeconomic status reduced risk of infection/morbidity which is contributed to reduce the prevalence of anemia?

Line 289-292. Can authors provide more evidence regarding to social beliefs and climatic conditions factors contribute to the reduction of anemia among women living in Tigray region? which is lover than in other regions.

Line 295-300. Why the marital status was discussed in these lines? According to result’s table, this study did not include this variable. It seem like authors raise unrelated ideas.

Line 300-303. Author showed the result of a study conducted in Indonesia, I am not sure how is related to the findings from this study? The explanation or reference from other studies should always come up after showing your findings.

Line 302-305. Authors need more discussion regarding to abortion, history of menstruation and electricity factors associated with anemia among women of reproductive age??? There is no explanation of these factors.

Do this study have any limitations?

Conclusion

Line 308-309. What is the meaning of percentage points?

Line 309-312. Please re-check the variables and re-phrases the sentences in these lines.

Line 312-314. How the policy-makers enhance the socio-demographic characteristics and basic infrastructure that could be contribute to reduce the prevalence of anemia? Authors should recommend the practical ideas which is related to the findings from this study?

Line 314-315. Please consider your recommendation with practical ideas in accordance with the findings from this study?

Line 315-316. Please consider your suggestion again on how to balanced the various factors among different state in Ethiopia? And how these factors prevent anemia in these states.

Reviewer #2: This is an interesting paper reporting anemia prevalence in Ethiopia. Should be checked for biostatistics .The paper could be shorten and comparison with prevalence of anemia in other African countries.

6. PLOS authors have the option to publish the peer review history of their article (what does this mean?). If published, this will include your full peer review and any attached files.

Reviewer #1: No

Reviewer #2: No

---

## [Author Response · Author response to Decision Letter 0]

15 Jun 2022

From ------------Berhan Tsegaye (Corresponding Author) 

To----------------Editor in chief 

Journal---------plose one

Date-----------26/05/2022A

Title --------Trend of anaemia among women reproductive age group in Ethiopia from 2005-2016: A Further analysis of Ethiopian demographic health survey.

Authors’ response for editor comment 

Dear editor 

Dear editor, we want to thank for the extensive revision of this paper which deals important public health issue. Based on the comments and questions we tried to revise the paper completely and extensively through several editing process. We revised our cover letter for free waiver for this paper as we are eligible for plose publication fee assistance program and our country is in G-1.we also attached the Ethical review letter for this study in the supplementary file. We are now ready for further improvement for this paper for readers if needed. If you need any clarification please contact the corresponding author. We have made the track change and clean manuscript in this revision. Besides, we answered the questions and comments of each reviewer as follows. 

Reviewer 1

Reviewer comment

General comment: Authors wrote very poor English, many of sentences were hard to understand. I strongly suggest the authors search for English proofreading services / checking by English native speakers to enhance English writing and reading. Data analysis was very poor which need to re-check, including the selected variables.

Authors’ response 

Dear reviewer, we would like to thank you for your extensive reviewing and give valuable comment. We accepted the comment and we corrected the language of the manuscript with extensive reviewing, editing and proof read through expert consultation also. In the current revision, we also are revising the tables and analysis of the variables. 

Reviewer comment 

This manuscript reported both the trend and factors associated with anaemia among women of reproductive age. I suggested revising the title of your manuscript to include factor associated with anaemia. Also, women of reproductive age means the group of women aged 15-49 years itself, so, it is better to remove word ‘group’ after women of reproductive age throughout this manuscript. In addition, in my understanding we usually call: Demographic “and” Health Survey (DSH), Please revise the correct word of this survey? If my comment is correct, you need to revise this word throughout manuscript.

Authors’ response 

Dear reviewer, once again we thank for the comment. It is relevant. We accepted the comment and corrected you suggestion in the current revision. In the current revision, we have made the following correction on the title indicated in bold: 

a. We add the phrase ‘factors associated’ with anaemia. 

b. We remove ‘group’ from the phrase which describe the study population ‘women in reproductive group’ as it is redundant.

c. We replaced description of ‘Ethiopian Demographic Health survey’ by ‘Ethiopian Demographic and Health survey’

Reviewer comment

 Abstract

i. Background. Please check the information regarding to anaemia is the most common cause of indirect morbidity?

Authors’ response 

Dear reviewer, according to your recommendation we have revised the sources for ‘most common cause of indirect morbidity’. Hence, we have found that anaemia is one of the 20% common cause ‘not the most’ cause of the indirect mortality. We accepted your concern and corrected it as ‘one of the common’ cause of indirect morbidity.

ii. Methods: Please revise these paragraph, many information is not related to the result section, particularly the odds ratio, 95% confidence interval are not showing in the result section at all.

Authors’ response 

Dear reviewer, once we want to ask apologizes and we wonder your patience as the paper lost integrity. We wrote rigorously the whole document in consecutive steps. We specifically wrote the results including COR and AOR consistently in different section of the paper. You can check it. 

Reviewer comment 

iii. Result. Please revise all sentences because in the main text did not show the results of the positive or negative association of several factors with anaemia.

Authors’ response 

Dear reviewer, we feel sorry for the inconvenience we made. We accepted the comment and revised it. We showed the direction and magnitude of the association for factors in the current revision clearly. 

Reviewer comment 

iv. Conclusion: Authors recommended several unrelated ideas which is not consistent with the findings from this study. Authors should revise this paragraph.

Authors’ response 

Dear reviewer, you are correct of course. We accepted the comment and revised the conclusion based only on the finding of the study. 

Reviewer’s comment 

The introduction or background section: It is better to reconstruct this paragraph by describing the situation of anaemia from global, regional (high-income/low-income countries), and country levels, respectively. Additionally, this study aimed to analyzed the trend of anaemia in Ethiopia, however, in this section is not describe about the trend of anaemia as the priority problem in this study. Moreover, this section is significantly long text, I strongly recommend the authors divided this section at least two paragraphs.

Authors’ response 

Dear reviewer, once we want to thank you. We wrote related sufficient context for the problem we raise in this revision coherently. We also wrote clearly and concisely by dividing paragraphs as you mention.

Reviewer comment 

Line 63. It is better to give the definition of anaemia for a common understanding for the readers.

Authors’ response 

Dear reviewer, we accepted your comment and correct it. 

Reviewer comment 

Line 66: Please revise this sentence with evidence regarding the cause of anaemia, because “it might be caused” is not a clear statement because anaemia can caused by infections, genetic conditions, etc. So, please clarify the cause of anaemia.

Authors’ response 

Dear reviewer, we have corrected and revised as your suggestion. We have corrected as the statement of fact rather than ‘hedge’ that express as possibility. 

Reviewers’ response 

Line 74-76. Can you give more information how this sentence related to anaemia among women of reproductive age, particularly what kind of extra foods and supplement needed during pregnancy and lactation? Also, what is the mechanism of the physiological changes, metabolism, growth of foetus and breast feeding during pregnancy and lactating affect to anaemia among these women?

Authors’ response 

Dear reviewers, the study population of this study are women of reproductive age which is from 15-49 in Ethiopian context. That means these women might be pregnant, lactating or neither of them. Hence, pregnancy and lactation can increase iron demand. For this reason, women are advised to take iron tablet (supplements). Besides, they are advised to take iron reach foods like: Teff, beef and egg, and to reduce drinks that reduce iron absorption like: Soft drinks and coffee.

Reviewers’ author 

Line 80-81. Physiological and nutritional change need to clarify for the completeness of sentence and can be merged with line number 74-76.

Authors’ response 

Dear reviewer, we have accepted and corrected your comment. we have merged the nutritional information together. 

Reviewers’ author 

Line 84. What is the meaning of grand multi-parity?

Authors’ response 

Dear reviewer, we mean for woman who gave birth more than five times are called ‘grand multipara’ while a woman delivered only once is called ‘primi-para’. For the sake of clear understanding of the audience in different profession, we wrote as the non-professional meaning of the term.

Reviewers’ comment 

Line 90-92. In my opinion, using the word “previous studies” instead of the word “former studies” might be better, which is commonly used in scientific report. In addition, I am not sure about the intention of author would like to talk about this sentence on how is related to anaemia research?

Authors’ response 

Dear reviewer, we thank again for you question. We accepted and incorporated the better explanation of previous studies than former let us make it clear. Furthermore, our intention for the sentence of line 90-92 is to show the gap on the specific population group (lactating and pregnant women) than the population of the current study (women in reproductive age group).

Reviewers’ comment 

What kind of the existing policies and programs which is addressed to anaemia problem among women of reproductive age in Ethiopia? What are the barriers of these policies and programs which could not reduce the prevalence of anaemia among these women in Ethiopia? Author might additional raise this important problem in this manuscript.

Authors’ response

Dear reviewer, the Ethiopian nutrition program and policy has put 3 main strategies towards anaemia: Iron-folic acid supplementation, food fortification and dietary diversification among women of reproductive age group in Ethiopia. Specifically, iron-folic acid tablets are usually given for pregnant women during their antenatal care visits. However, compliance is generally low with great regional variations, 3.5% to 76% and some of the strategies proposed such as food fortification have not yet implemented.

Reviewers’ comment 

Line 99-101. Please revise this sentence: First point: This study “aimed” to analyse the trend and factors associated with anaemia among…., Second point: If you use the word women of reproductive age, you need to use the same wording throughout manuscript to avoid the misunderstanding. Third point: the national data using to analyse in this study should be clearly with exact name and years.

Authors’ response

Dear reviewer, we have received these important comments and modify accordingly. 

Methods section 

Reviewers’ comment 

General comment on study setting section. This paragraph is quite complicated in term of sentence order and unrelated ideas. This paragraph should be clearly provided information regarding to settings and locations where the data were collected and locations in which the study was carried out, including the country, region, and city. Also, authors could provide other information about settings and locations that could have affected a study’s external validity such as social, economic, cultural environment and other special aspect of study settings, if applicable. In addition, I would like to suggest the authors to provide the decision trees that would be innovative to explore such data (merely a thought) in order to show the readers on how the sample were selected and excluded at each stage.

Authors’ response 

Dear reviewer, we read and agree on your comment. Hence, we made revision based on it. We also prepared a figure which shows the schematic presentation of the sampling technique.

Reviewer comment 

Line 107-109. Please revise this sentence started from About 20%..... until in rural areas by giving the exact numbers/percentages of people living in rural and urban areas. Nearly 80% is not exact number of 80%.

Authors’ response 

Dear reviewer, we have accepted and reviewed the comment. We have corrected in this revision.

Reviewer comment 

Line number 109-110. 1st point, I am not sure that we can say 4.6 individuals/persons or what is the meaning of 4.6 persons? 2nd point, this study used three different dataset of 2005, 2011 and 2016, but in this sentence mentioned only the survey conducted in 2016.

Authors’ response 

Dear reviewer, we accepted and corrected your comment. Actually, the average household in both in 2011 and 2016 was 4.6 persons/household. But, the average household size in 2005 was 5. In the revision, we exclusively describe household size for each DHS. 

Reviewer comment 

Line 111-137. Data source/study design, study population and sampling technique. The text of this section can be divided in three paragraphs separately under this sub-heading.

- Data source should be clearly mentioned the source of datasets using/analyzing in this study, year of conducting these surveys, type of data (primary/secondary datasets?) along with providing information about these surveys.

- Study population.

- Sampling technique. In my understanding the DHS survey used the multi-stage cluster sampling technique. Authors can re-check in the report of each original surveys, and please provide the related information into this section.

Authors’ response 

Dear reviewer, we have corrected according to your advice. 

Reviewer comment 

- DHS surveys typically provide altitude and smoking-adjusted levels of hemoglobin. In your study, I truly believe that smoking is not a concern; however, altitude might be. During data collection and building databases, some factors were used to adjust the levels of hemoglobin?

Authors’ response 

Dear reviewers, we want to thank you for your nice comment. We have excluded pregnant women and smoker women whom information about smoking was collected as these factors misclassify cases and cannot be merged and analyzed with women of reproductive age group. Therefore, these women should be disaggregated and analyzed. However, haemoglobin was adjusted by consideration of Altitude. The calculation was as follows:

Calculation 

At the time of creation of a recode file, an adjustment of the haemoglobin count is made for altitude. Rather than change the cut off points, the effective haemoglobin count is lowered as altitude increases, since oxygen is less available. The adjustment is made with the following formulas:

Adjust 

 adjHb if adjust > 0

where adjust is the amount of the adjustment, alt is altitude in 1,000 feet (converted from meters by dividing by 1,000 and multiplying by 3.3), adjHb is the adjusted haemoglobin level, and Hb is the measured haemoglobin level in grams per decilitre. No adjustment is made for altitudes below 1,000 meters. Both the adjusted and unadjusted haemoglobin counts are included in the recode files.

Similarly, an adjustment is made for women who smoke (if information was collected). The adjustment is to be made in accordance with the following table:

Cigarettes Smoked Adjust Hb (g/dl) concentration by

Less than 10 per day No adjustment

10-19 per day -0.3

20-39 per day -0.5

40 or more per day -0.7

Unknown quantity or non-cigarettes smoking -0.3

Reviewer’s comment 

Line 116. Authors should provide the full text for EDHS as the first time state follow by the abbreviation in blanket. Also, this study used three cross-sectional data of 2005, 2011, and can authors provide the reason on this matter for clarification?

Authors’ response 

Dear reviewer, we have corrected according to your advice and 2016 but the sampling frame mentioned only in 2016. We apologize for ignorance of the sampling frame for 2005 and 2011 sampling frame. We mentioned for all surveys in this revision.

Reviewer’s comment

Line 165. Non-governmental organizations.

Authors’ response 

Dear reviewer, we want to describe the stakeholders which participate in DHS like non-governmental organization but since the detail of the methodology is found in the report of each EDHS, we want remove such texts to shorten the manuscript by making citation. 

Reviewer’s comment

Line 126. What is the different between cluster and EA in this sentence?

Authors’ response 

Dear reviewer, enumeration areas and clusters are similar words in this study so that we can use them interchangeably. 

Reviewer’s comment

Line 127-134. 17,067 households had women of reproductive age. Can authors clarify why all of these women were not included? What is the eligible selection criterion?

Authors’ response 

Dear reviewer, all households were not occupied by women of reproductive age during data collection time.

Reviewers’ response

Study variables

General comment for this section. This section is absolutely complicated with unnecessary information. It is better to separate this section into two paragraphs to show the dependent and independent variables separately, and under the independent variables can be divided in two categorization of socio-demographic characteristics of the participants follow by reproductive characteristics. Furthermore, due to this section describe specifically for the study variables, therefore, the information regarding to data collection and its process should be mention in above section of the data source/study design, study population and sampling technique.

Authors’ response 

Dear reviewer, we thank again. We accepted the comment and correct it.

Reviewer comment 

Many independent variables were not mentioned in this section, but showed in result’s tables such as work status, residence, region, and frequency of reading new papers, frequency of watching TV, Abortion, electricity, telephone, menstruated in the last six months. On the other hands, many independent variables mentioned in this section, but not showed in result’s tables such as husband educational status, age at first pregnancy, number of birth, number of antenatal visit, and current pregnancy. Authors must re-organize it clearly, and each variable must show its categorization, for example: education (uneducated vs educated).

Authors’ response

Dear reviewer, we have made correction extensively in this revision. Hence, we describe these variables consistently in different part of the paper to avoid ambiguity. Especially, variables in the variables section and tables. Furthermore, we have described the variables organization and categorization.

Reviewer comment 

 Line 139-144. Authors provide two times regarding the definition of anaemia among non-pregnant women; authors can select one of them. Also, please provide full definition regarding of anaemia among pregnant and non-pregnant women because its definitions used the different cut point for mild, moderate and severe anaemia.

Authors’ response

Dear reviewer, we want to thank for your critical view. Since our study population were women of reproductive age, we selected them and remove the information about pregnant women. Besides, we critically define anaemia among women of reproductive age.

Reviewer comment 

Line 180. Please give the full text of CSA before it first statement of its abbreviation.

Authors’ response

Dear reviewer, we have accepted and corrected the comment. 

Reviewer comment 

Data management and statistical analysis.

In statistical section, please refer to any post-hoc corrections to correct for multiple comparisons during your statistical analyses. If these were not performed please justify the reasons. Additionally, in your statistical analyses, please state whether you accounted for clustering by Kebeles or region? For example, did you consider using multilevel models?

Author’s response

Dear reviewer, we did not use post-hoc correction for correcting multiple comparisons. Although Post hoc tests do a great job of controlling the family-wise error rate, but the trade-off is that they reduce the statistical power of the comparisons. This is because the only way to lower the family-wise error rate is to use a lower significance level for all of the individual comparisons. Furthermore, we did not perform multi-level model (mixed effect model) analysis, as we did not assume multi-level factors are associated with anaemia. Moreover, our data did not allow multi-level analysis as ICC >0.05. However, we performed weighing and complex 

Reviewer comment 

Line 188. How many datasets were analyzed for this study? Three or four? And it should be mentioned the year of each datasets!!data analysis for accounting the sampling bias. 

Author’s response

Dear reviewer, we have utilized 3 datasets (EDHS 2005, EDHS 2011 and EDHS 2016) for women data. We have made sampling weight and complex analysis for each survey. 

Reviewer comment 

Line 189-190. Authors must understand this section which must provide only the statistical tests used/analyzed by you.

Authors’ response 

Dear reviewer let us clarify on the sampling weight and complex sampling issue. DHS is not only simple raw data from scratch. It is designed and analyzed to some extent like descriptive report is given and the way how to deal with the data to the end of analysis for the given topic has been elaborated in the report and videos are prepared by owner of the data for maximum utilization of the data by the end user (authors). That means the data can be compiled in many forms and given for analysis clarifying steps and codes how to do. The data can be provided in different software packages. We receive it with SPSS software so that we analyzed following the instruction for each dataset for a given population. Consequently, we can get the result finally. Therefore, we analyzed by ourselves following instruction of the steps of the data owner.

NB: some variables were re-categorized or recoded based on the aim of the study and LR in this study. 

Reviewer comment

Line 191. Are authors used the sampling weight for all analyzes?

Authors’ response 

Dear reviewer, we adjust the each data set (EDHS 2005,2011 and 2016) using the weighting variable. 

Reviewer comment

Line 193-194. Which statistical tests whether chi-square or logistic used to analyzed the date?

Authors’ response 

Dear reviewer, we have used both tests. We used chi-square to test the presence of association only. However, since chi-square test cannot show magnitude and direction of the association. We further analysis in logistic regression. We showed the result of logistic analysis with table. 

Reviewer comment

Line 194: In which criteria those authors selected variables into multivariable logistic regression? And how do you treat with multicollinearity problem? These can be strengthened with a reference at least.

Authors’ response

Dear reviewer, we use p-value less than 0.25 for variable eligibility criteria for multi-variable logistic regression analysis. Multicollinearity tests were performed to check the presence of correlations among explanatory factors. We computed the variance inflation factor (VIF) for each predictor variable by doing a logistic regression of each predictor on all the other predictors; in each case we obtained VIF within the range of recommended cut of points

Reviewer comment

Line 195-196. I did not see the odds ratio and 95% confidence interval in the result’s tables.

Authors’ comment 

Dear reviewer, we mentioned COR, AOR and their 95%CI. 

Reviewer comment

Line 196-197. Which number of p-value considers as statistical significant 0.25 or 0.05 and please cited a reference?

Authors’ comment 

Dear author, we used p-value less than 0.25 for binary logistic regression analysis and 0.05 in multi-variable logistic regression analysis as the level of significance. 

Reviewer comment

Line 198-200. Please rephrase this sentence, I don't understand. Also, what is the different between univariate and bivariate analyses?

Authors’ response

Dear reviewer, we thank again for the comment. Let us clarify the issue, univariate analysis means simply descriptive analysis which consist only variable. Bivariate means binary logistic regression analysis. Hence, we described them separately for clear explanation as ‘descriptive’ data analysis and ‘binary logistic regression analysis’. 

Reviewer comment 

Line 203-208. Ethical consideration: Understandable, the secondary data analysis is not required the double ethical approval; however, authors should provide additional details regarding participant consent as well as which organization/agencies provided ethical approval for these surveys. Please ensure that you have specified what type you obtained (for instance, written or verbal, and if verbal, how it was documented and witnessed). If the need for consent was waived by the ethics committee, please include this information.

Authors’ response 

Dear reviewer, Ethics approval and participant consent were not necessary as this study involved the use of a previously-published de-identified database by Central Statistical Agency of Ethiopia. 

Reviewer comment 

Result

Can authors show the number of participant in result’s tables 1 and 2 for each categories of each variables follow by its percentages?

Authors’ response

Dear reviewer, we can do that but the information of the table become overburden so that we only write the weighted percentage for each variable level or cross tabulation. We also recommended this issue from other paper we publish to ignore double reporting of frequencies both in number and their percentage as it reduce the quality of paper. 

Reviewer comment 

Many independent variables were not showed in result’s tables 1 and 2 but mentioned in method section such as husband education, marital status, age at first pregnancy, number of birth, number of antenatal visit, current pregnancy. Authors need to re-check for clarification!!!

Authors’ response

Dear reviewer, we received and corrected the comment. 

Reviewer comment 

Authors did not show the result of logistic regression analyses with the result of odds ratio and 95% confidence interval?

Authors’ response

Dear reviewer, we received and corrected the comment. 

Reviewer comment 

Why the menstruated information was asked in the last six weeks? Because the menstruation were usually happened as monthly basis.

Authors’ response 

Dear reviewer, the level of anaemia depends on amount of blood loss, duration of blood loss, and cycle of menstruation. It is clear that when the women bleed for longer months, they tend to be anaemic.

 Reviewer comment

Line 210. Please revise this sub-heading. The correct word is “characteristics” and please re-checks this word in other section as well.

Authors’ response

Dear reviewer we have accepted and corrected the comment. 

Reviewer comment 

Line 211. Mean age of the participants was 28, is ±9.6 years the standard deviation? Please provide clearly in such phrase.

Authors’ response 

Dear reviewer we have accepted and corrected the comment. 

Reviewer comment

Line 211-214. Majority mean more than half. Please revise the related sentences, along with providing full description of result of selected variables from each year of surveys 2005, 2011, and 2016, respectively. Due to this study used three different datasets from different years, authors should be careful to provide the clear description.

Authors’ response 

Dear reviewer we have accepted and corrected the comment. 

Reviewer comment 

Line 216-217. In result section, authors should provide the exact number of percentage of each selected variable. The least proportion is not a common used in scientific report.

Table 1. (i) Please revise the name of this table, (ii) showing two p-value in a table can cause misunderstanding, other might select one of them and need to explain in statistical analyses section. (iii) age category is not in the similar frequency, for example: 15-19 years is within 5 years but other category of 20-29 years is within 10 years, do you have any reference to strengthen this categorization?

Authors’ comment 

Dear reviewer, we accepted and corrected your comment for removing confusion for our reader of the paper. Furthermore, we corrected the age range actually this was the editing problem. 

Reviewer comment

Line 226-230. I recommend to re-interpret the results of the table 2?? A swell as revising the name of the table 2.

Authors’ comment 

Dear reviewer, we totally accepted your comment and revising the result in the table and its labelling 

Reviewer comment

Line 237-249. It is better to show the figures before or after the result’s interpretation. Authors should interpreted each figure with a separate texts.

Authors’ comment 

Dear reviewer, we accepted and corrected the comment. 

Reviewer comment

Line 239-240. What is the different between 47.7% and 47.7 percentage points?

Authors’ comment 

Dear reviewer let us clarify this issue. 47.7% is simply the prevalence (proportion out of 100). However, percentage point means it is the difference in prevalence or the increase in prevalence from one certain level to the other (next). 

Reviewer comment 

Line 250-256. I am not sure that the result in this section came from which tables? The results are not show!!!!

Authors’ response

Dear reviewer, we want to ask apologize for the missing tables. We corrected and supplied in the current revision as table 3&4. 

Reviewer comment 

Discussions:

In your discussions, please take care to avoid statements implying causality from correlational research. For example, avoid the use of terms such as "increased/reduced risk", "influenced by", “likely to” or “resulted in." Instead, consistently use terms such as "associated with" or "associations."

Authors’ response 

Dear reviewer, we have got your idea. Your comments are well accepted. 

Reviewer comment 

Line 261. Anaemia vs. anaemia are American and British English, please select one of them for whole manuscript.

Authors’ response 

Dear reviewer, we have selected the second (British) word and made consistent throughout the document currently.

Reviewer comment 

Line 261-265. Authors don’t need to give the definition of anaemia in this section which is already mentioned in the background section.

Authors’ response 

Dear reviewer, we have accepted the comments and corrected it. 

Reviewer comment 

Line 266. In statistic, to show the full number like 68% must follow by dots zero as 68.0%, authors should revise other full numbers as well.

Authors’ response 

Dear reviewer, we have accepted the comments and corrected it. 

Reviewer comment 

Line 265-277. It is good to compare the prevalence with other countries; however, the author should try to find the evidence to answer why the trend of prevalence of anaemia was significantly reduced from 2005 to 2011, but increased from 2011 to 2016 in Ethiopia? This should be the main discussion regarding to the findings from this study. Had it any policy/intervention or event which influenced to these unstable trends? The comparison of prevalence of anemia among children and pregnant women with women of reproductive age might lead to unrelated ideas, because the study participants are different. Also, authors should provide the year of the references’ cited for example in line 275-277, recent national DHS and other surveys were conducted in which years? is that only one reference for respective countries?

Authors’ response 

Dear reviewer, we have accepted the comments and corrected it. The study population were women of reproductive age group one of the vulnerable group for anaemia. We correct those unrelated ideas and cite the specific year for the data and references. 

Reviewer comment 

Line 277-278. What is the reduction of anemia among women of reproductive age important for child health?

Authors’ response 

Dear reviewer, the anaemia level of women is highly associated with intrauterine life of the foetus and child. For example, still birth, abortion, low birth weight and preterm labor. Furthermore, infants are dependent of their mothers’ iron store during early childhood during lactation. This explanation is supported by previous studies. 

Reviewer comment 

Line 228-280. What is the meaning of this sentence “anaemia prevalence remains high and haemoglobin levels remain low in the low income countries”?.

Authors’ response 

Dear reviewer, we want to ask for those inconsistent sentences. we have accepted and corrected currently. 

Reviewer comment 

Line 280-281. “If these trends are continued, the likelihood of reducing anemia by half from 2011 levels by 2025 in all regions among reproductive age women”. This prediction might be happened and might not be happened as well, the scientific research should come up with the fact and real evidence.

Reviewer comment 

Line 282-283. What is the group of lower wealth indexes? According to tables’ result showed five categories from poorest to richest group? Which group is the reference group?.

Reviewer comment 

Line 283-285. Can authors provide more evidence regarding to women from high socioeconomic status reduced risk of infection/morbidity which is contributed to reduce the prevalence of anaemia?

Reviewer comment 

Line 289-292. Can authors provide more evidence regarding to social beliefs and climatic conditions factors contribute to the reduction of anaemia among women living in Tigray region? This is lower than in other regions.

Reviewer comment 

Line 295-300. Why the marital status was discussed in these lines? According to result’s table, this study did not include this variable. It seem like authors raise unrelated ideas.

Authors’ response 

Dear reviewer, we accepted your comment. Marital status is one of the main factors associated with anaemia among women. We accepted that the result table was incomplete to show these factors. We have corrected in the current revision.

Reviewer comment 

Line 300-303. Author showed the result of a study conducted in Indonesia, I am not sure how is related to the findings from this study? The explanation or reference from other studies should always come up after showing your findings.

Authors’ response 

Dear reviewer, we have accepted your concern and revised it. 

Reviewer comment 

Line 302-305. Authors need more discussion regarding to abortion, history of menstruation and electricity factors associated with anemia among women of reproductive age??? There is no explanation of these factors.

Authors’ response 

Dear reviewer, we have described adequately in the current revision. 

Reviewer comment 

Do this study have any limitations?

Authors’ response 

Dear reviewer, we thank you for remind us. This study had strengths and limitations. We included them in the current revision. 

Reviewer comment 

Conclusion

Line 308-309. What is the meaning of percentage points?

Authors’ response 

Dear reviewer, we have described that it is the difference of two percentage/prevalence/ 

Reviewer comment 

Line 309-312. Please re-check the variables and re-phrases the sentences in these lines.

Authors’ response 

Dear reviewer, we accepted your advice and revised it. 

Reviewer comment 

Line 312-314. How the policy-makers enhance the socio-demographic characteristics and basic infrastructure that could be contribute to reduce the prevalence of anemia? Authors should recommend the practical ideas which are related to the findings from this study?

Authors’ response 

Dear reviewer, we accepted your advice and revised it. 

Reviewer comment 

Line 314-315. Please consider your recommendation with practical ideas in accordance with the findings from this study?

Authors’ response 

Dear reviewer, we accepted your advice and revised it. 

Reviewer comment 

Line 315-316. Please consider your suggestion again on how to balance the various factors among different state in Ethiopia? And how these factors prevent anaemia in these states.

Authors’ response 

Dear reviewer, we accepted your advice and revised it. 

Reviewer #2: comment 

This is an interesting paper reporting anemia prevalence in Ethiopia. Should be checked for biostatistics .The paper could be shorten and comparison with prevalence of anemia in other African countries.

Author’s response 

Dear reviewer, we want to thank you for your nice comment. We have tried as much as possible to shorten the paper compare the findings with other comparative African countries and we also consult the biostatician for the statistics in our paper.

---

## [Decision Letter · Decision Letter 1]

22 Sep 2022

PONE-D-21-33298R1Trend and factors associated with anemia among women reproductive age in Ethiopia: A Further analysis of Ethiopian Demographic and Health Survey from 2005-2016.PLOS ONE

Dear Dr. Berhan Tsegaye,

Thank you for submitting your manuscript to PLOS ONE. After careful consideration, we feel that it has merit but does not fully meet PLOS ONE’s publication criteria as it currently stands. Therefore, we invite you to submit a revised version of the manuscript that addresses the points raised during the review process.

ACADEMIC EDITOR:   dear author ,  there are still grammatical errors in the manuscript . please correct them also review your statistical analysis as suggested by reviewer. 

Please submit your revised manuscript by 5th Oct 2022. If you will need more time than this to complete your revisions, please reply to this message or contact the journal office at plosone@plos.org. Please include the following items when submitting your revised manuscript:A rebuttal letter that responds to each point raised by the academic editor and reviewer(s). You should upload this letter as a separate file labeled 'Response to Reviewers'.A marked-up copy of your manuscript that highlights changes made to the original version. You should upload this as a separate file labeled 'Revised Manuscript with Track Changes'.An unmarked version of your revised paper without tracked changes. You should upload this as a separate file labeled 'Manuscript'.If applicable, we recommend that you deposit your laboratory protocols in protocols.io to enhance the reproducibility of your results. Protocols.io assigns your protocol its own identifier (DOI) so that it can be cited independently in the future. For instructions see: https://journals.plos.org/plosone/s/submission-guidelines#loc-laboratory-protocols. Additionally, PLOS ONE offers an option for publishing peer-reviewed Lab Protocol articles, which describe protocols hosted on protocols.io. Read more information on sharing protocols at https://plos.org/protocols?utm_medium=editorial-email&utm_source=authorletters&utm_campaign=protocols.

We look forward to receiving your revised manuscript.

Kind regards,

Sidrah Nausheen, FCPS

Academic Editor

PLOS ONE

Journal Requirements:

Reviewers' comments:

Reviewer's Responses to Questions

**Comments to the Author**

1. If the authors have adequately addressed your comments raised in a previous round of review and you feel that this manuscript is now acceptable for publication, you may indicate that here to bypass the “Comments to the Author” section, enter your conflict of interest statement in the “Confidential to Editor” section, and submit your "Accept" recommendation.

Reviewer #2: All comments have been addressed

2. Is the manuscript technically sound, and do the data support the conclusions?

Reviewer #2: Partly

3. Has the statistical analysis been performed appropriately and rigorously? 

Reviewer #2: N/A

4. Have the authors made all data underlying the findings in their manuscript fully available?

Reviewer #2: Yes

5. Is the manuscript presented in an intelligible fashion and written in standard English?

Reviewer #2: Yes

6. Review Comments to the Author

Reviewer #2: The paper could be published after statistical review.Although not very novel but is an important attempt to study anemia

in developing country

7. PLOS authors have the option to publish the peer review history of their article (what does this mean?). If published, this will include your full peer review and any attached files.

Reviewer #2: No

---

## [Author Response · Author response to Decision Letter 1]

25 Oct 2022

From: Berhan Tsegaye

Date: 25/10/2022

To: The Editor

Journal Name: Plose one

Re-submission of a manuscript 

Dear Editor/reviewer/, I am Berhan Tsegaye, on behalf of correspondent authors submits the following manuscript for publication. ‘Trend of anemia among women reproductive age in Ethiopia from 2005-2016: A Further analysis of Ethiopian demographic health survey’. 

Dear editor, we are asked to correct the language and statistical analysis in this revision round. 

Hence, we have revised, edited the manuscript again and again. Furthermore, we have made some statistical revision which can not affect the previous result. We have corrected the manuscript though many consecutive rounds so some revision including statistical revision might not be indicated in the track change. Furthermore, as we are from Ethiopia, we cannot pay all cost of the publication and we are eligible for ‘PLOS Publication Fee Assistance Program’. We want to remind this ahead of time. Once again, we would like to thank all the reviewers and editors for their comment and contributions. We are also ready to take any suggestions and comments further if it is necessary. 

Corresponding Author

Berhan Tsegaye 

Email – birieman67@gmail.com

---

## [Decision Letter · Decision Letter 2]

7 Dec 2022

PONE-D-21-33298R2Trend and factors associated with anemia among women reproductive age in Ethiopia: a multivariate decomposition analysis of Ethiopian Demographic and Health survey.PLOS ONE

Dear Dr. Tsegaye,

Thank you for submitting your manuscript to PLOS ONE. After careful consideration, we feel that it has merit but does not fully meet PLOS ONE’s publication criteria as it currently stands. Therefore, we invite you to submit a revised version of the manuscript that addresses the points raised during the review process. ==============================Dear author

there are multiple grammatical mistakes which need to be corrected

you said menstruation in last six months is associated with anemia. every woman has menstruation, but all are not anemic. so please change you statement and write heavy menstruation is associated with anemia.

reading news paper has protective effect in your study then why using mobile phone is not protective . mobile phone and social media gives you more information so please correct line 510 and 511.

why are you associating anemia with electricity ?? it does not make sense . either remove it or give logical interpretation

We look forward to receiving your revised manuscript.

Kind regards,

Sidrah Nausheen, FCPS

Academic Editor

PLOS ONE

Journal Requirements:

Additional Editor Comments:

dear author

there are multiple grammatical mistakes which need to be corrected

you said menstruation in last six months is associated with anemia. every woman has menstruation, but all are not anemic. so please change you statement and write heavy menstruation is associated with anemia.

reading news paper has protective effect in your study then why using mobile phone is not protective . mobile phone and social media gives you more information so please correct line 510 and 511.

why are you associating anemia with electricity ?? it does not make sense . either remove it or give logical interpretation

Reviewers' comments:

Reviewer's Responses to Questions

**Comments to the Author**

1. If the authors have adequately addressed your comments raised in a previous round of review and you feel that this manuscript is now acceptable for publication, you may indicate that here to bypass the “Comments to the Author” section, enter your conflict of interest statement in the “Confidential to Editor” section, and submit your "Accept" recommendation.

Reviewer #2: All comments have been addressed

2. Is the manuscript technically sound, and do the data support the conclusions?

Reviewer #2: Yes

3. Has the statistical analysis been performed appropriately and rigorously? 

Reviewer #2: Yes

4. Have the authors made all data underlying the findings in their manuscript fully available?

Reviewer #2: Yes

5. Is the manuscript presented in an intelligible fashion and written in standard English?

Reviewer #2: Yes

6. Review Comments to the Author

Reviewer #2: The authors have responded to comments and the paper now is acceptable.

The paper is acceptable for publication

7. PLOS authors have the option to publish the peer review history of their article (what does this mean?). If published, this will include your full peer review and any attached files.

Reviewer #2: No

---

## [Author Response · Author response to Decision Letter 2]

19 Dec 2022

From: corresponding author

To: Editor in chief 

Submission ID: PONE-D-21-33298R2

Journal: plose one 

Title: Trend and factors associated with anaemia among women reproductive age in Ethiopia: a multivariate decomposition analysis of Ethiopian Demographic and Health survey.

General comment of the editor

Dear author

There are multiple grammatical mistakes which need to be corrected

you said menstruation in last six months is associated with anemia. Every woman has menstruation, but all are not anemic. So, please change you statement and write heavy menstruation is associated with anemia.

Authors’ response 

We want to thank the editor for his/her valuable comment for general correction and editorial problem. We accepted and corrected the language problems (grammatical errors, misplaced modifier tense etc.) throughout the document in the current revision. We have tried our effort to make our paper readable for our readers. 

Editor’s specific comment 

Reading newspaper has protective effect in your study then why using mobile phone is not protective. Mobile phone and social media gives you more information so please correct line 510 and 511.

Authors’ response

We want the reviewer to thank for this comment. We wrongly interpreted the correct finding in the table which is editorial problem. We have corrected it. Social media had great impact in prevention of anemia.

Editor’s specific comment 

Why are you associating anaemia with electricity?? It does not make sense. Either remove it or give logical interpretation

Authors’ response

We thank the reviewer for this comment. We have given the logical interpretation with evidence.

---

## [Editor Report · Decision Letter 3]

6 Jan 2023

Trend and factors associated with anemia among women reproductive age in Ethiopia: a multivariate decomposition analysis of Ethiopian Demographic and Health survey.

PONE-D-21-33298R3

Dear Dr. Berhan Tsegaye,

We’re pleased to inform you that your manuscript has been judged scientifically suitable for publication and will be formally accepted for publication once it meets all outstanding technical requirements.

Kind regards,

Sidrah Nausheen, FCPS

Academic Editor

PLOS ONE
---

## [Editor Report · Acceptance letter]

12 Jan 2023

PONE-D-21-33298R3 

Trend and factors associated with anemia among women reproductive age in Ethiopia: a multivariate decomposition analysis of Ethiopian Demographic and Health survey. 

Dear Dr. Tsegaye Negash:

I'm pleased to inform you that your manuscript has been deemed suitable for publication in PLOS ONE. Congratulations! Your manuscript is now with our production department. 

Kind regards, 

on behalf of

Dr. Sidrah Nausheen 

Academic Editor

PLOS ONE